# Extreme Weather Nowcasting via Local Precipitation Pattern Prediction

**Changhoon Song**[*] **& Teng-Yuan Chang**[*]
Research Institute of Mathematics
Seoul National University
Seoul, Republic of Korea
`{changhoon.song93,tony890048}@snu.ac.kr`

**Youngjoon Hong**[†]
Department of Mathematical Sciences & Research Institute of Mathematics
Seoul National University
Center for AI and Natural Sciences, Korea Institute for Advanced Study (KIAS)
Seoul, Republic of Korea
`hongyj@snu.ac.kr`

## Abstract

Accurate forecasting of extreme weather events such as heavy rainfall or storms is critical for risk management and disaster mitigation. Although high-resolution radar observations have spurred extensive research on nowcasting models, precipitation nowcasting remains particularly challenging due to pronounced spatial locality, intricate fine-scale rainfall structures, and variability in forecasting horizons. While recent diffusion-based generative ensembles show promising results, they are computationally expensive and unsuitable for real-time applications. In contrast, deterministic models are computationally efficient but remain biased toward normal rainfall. Furthermore, the benchmark datasets commonly used in prior studies are themselves skewed–either dominated by ordinary rainfall events or restricted to extreme rainfall episodes–thereby hindering general applicability in real-world settings. In this paper, we propose exPreCast, an efficient deterministic framework for generating finely detailed radar forecasts, and introduce a newly constructed balanced radar dataset from the Korea Meteorological Administration (KMA), which encompasses both ordinary precipitation and extreme events. Our model integrates local spatiotemporal attention, a texture-preserving cubic dual upsampling decoder, and a temporal extractor to flexibly adjust forecasting horizons. Experiments on established benchmarks (SEVIR and MeteoNet) as well as on the balanced KMA dataset demonstrate that our approach achieves state-of-the-art performance, delivering accurate and reliable nowcasts across both normal and extreme rainfall regimes. The code is publicly available at `https://github.com/tony890048/exPreCast`.

## 1 Introduction

Rainfall is essential for life and ecosystems, but excessive or intense precipitation often lead to severe natural disasters such as floods or landslides. With climate change, extreme rainfall events are becoming more frequent and intense, further underscoring the importance of accurate precipitation forecasting. To anticipate such events and prepare following risk, precipitation forecasting has been a central subject of meteorological study. While a variety of numerical weather prediction (NWP) models have been developed, it still faces limitations: the high computational cost that makes real-time forecasting challenging, limited spatial resolution that restricts the ability to capture local events, or physical parameterization imperfect to represent complex earth system.

---

[*]Equal contribution authors.
[†]Corresponding author.

Motivated by advances of deep learning in various scientific domains, recent studies have aimed to develop a data-driven model that directly learns spatiotemporal patterns from observations. In particular, for precipitation forecasting, radar-based forecasting model have been proposed as it provides high-resolution and real-time data. Those models have demonstrated performance on benchmark datasets such as SEVIR or MeteoNet. However, SEVIR is heavily biased to storm events, while MeteoNet consists of mostly normal rains. Consequently, it remains difficult to evaluate generalization of a model to both ordinary precipitation throughout year and rare extreme events.

In this work, we introduce a deterministic system for radar-based local rainfall nowcasting with a focus on extreme precipitation events, and a new dataset that contains both normal and heavy rains. Specifically, we adopt a Video Swin Transformer(Liu et al., 2022) to focus on local patterns, with *Cubic Dual-Upsample (CDU)* decoder to enhance texture fidelity. In addition, the decoder upscales temporal dimension to capture more diverse dynamics. Following *Temporal Extractor* (TE) compress those features to adjust forecasting time horizon without significantly increasing computational cost.

We design the overall architecture to address properties of weather forecasting distinct from usual video prediction. Fisrt, since rainfalls are determined by local meteorological phenomena, we restrict attention blocks using shifted window so that the feature learns local patterns rather than global relation. In addition, we introduce CDU that replaces the traditional upsampling to preserve high-frequency features. Although linear or pixel-shuffle upsampling are known to be efficient for various computer vision task, we observed that linear upsampling strongly smooths out the prediction and removes small area with higher intensity, where could be regarded as a noise in video prediction. On the other hand, pixel shuffling preserved high-frequency feature in radar but generate unrealistic artifacts. We therefore introduce CDU block that fuses trilinear interpolation and 3D pixel-shuffling to remove the artifacts while not compromising the details, thereby enabling to predict extreme events on small regime. Lastly, since CDU expands temporal dimension to learn flexible dynamics while address spatial details or radar data, TE adjust the time steps between outputs and extract dynamics.

To assess performance across a wide spectrum of rainfall events, we collect radar data from *Korea Meteorological Administration (KMA)*. Owing to the unique meteorological characteristics of South Korea–where seasonal monsoons and typhoons bring intense rainfall in summer, while lighter precipitation occurs regularly in other seasons–the dataset provides a balanced distribution of rainfall intensities. We evaluate our model leveraging KMA, SEVIR, and MeteoNet dataset, and achieve state-of-the-art performance.

Our main contributions are summarized as follow:

- We propose *exPreCast*, a deterministic radar-based forecasting architecture that achieves state-of-the-art accuracy across KMA, SEVIR, and MeteoNet, while maintaining substantially lower computational cost.

- We introduce the CDU block, which fuses trilinear interpolation with 3D pixel shuffle to mitigate checkerboard artifacts and retain small-scale high-intensity rainfall patterns. CDU consistently improves CSI scores, especially with pooling, demonstrating enhanced local-pattern fidelity.

- We construct a new large-scale dataset encompassing both ordinary and extreme precipitation events, offering more balanced meteorological coverage than existing benchmarks and enabling rigorous evaluation of generalization.

## 2 RELATED WORKS

**Spatiotemporal processing**   Weather forecasting can be formulated as a spatiotemporal sequence prediction, using past observation to forecast outputs in future. Shi et al. (2015) combined recurrent and convolutional layers in the ConvLSTM to capture spatiotemporal correlations. Based on Fourier Neural Operator (FNO) proposed by Li et al., Pathak et al. (2022) introduced Adaptive Fourier Neural Operator (AFNO) and FourCastNet, a large-scale weather forecasting system. In parallel, computer vision research advanced transformer-based architectures for sequential image; Liu et al. (2022) presented the Video Swin Transformer, which leverages hierarchical shifted-window attention to efficiently model local and global context in a video sequence. We adopt this as the backbone and extend with domain-specific modules tailored to radar-based rainfall nowcasting.

**Radar based nowcasting**  Beyond generic spatiotemporal architectures, a number of models have been developed specifically for radar-based precipitation nowcasting. For instance, Gao et al. (2022b) proposed EarthFormer, a space-time attention to learn local-global information. More recently, Gao et al. (2023) and Gong et al. (2024) have used a diffusion-based approach to predict fine features and leverage ensemble methods. Although they have shown state-of-the-art (SOTA) performance, the diffusion models require much higher computational costs than deterministic counterparts. Lin et al. (2025) separate networks that learn phase and amplitude, then combine those features in mixer module. It outperforms other baselines such as Earthformer, NowcastNet, and DiffCast on various benchmarks, including SEVIR and MeteoNet, but the datasets are downscaled to greater than 3 times due to computational limitations. In contrast, we develop a deterministic model that achieves SOTA results on SEVIR, MeteoNet, and KMA datasets. Our model requires significantly less computational cost and the performance is evaluated at the original resolution of SEVIR and MeteoNet. Moreover, it demonstrates consistently superior performance in forecasting extreme rainfall events.

**Radar datasets**  The development of deep learning models has followed the large-scale benchmark datasets. Veillette et al. (2020) suggested the Storm Event Image (SEVIR) dataset, which is one of the most commonly used benchmarks for evaluating radar-based precipitation forecasting models. SEVIR contains sequences collected across the United States and provides radar-based observations such as NEXRAD vertically integrated liquid (VIL), satellite imagery, and lightning. As it collects radar data from severe convective storms, SEVIR is biased to heavy rains but less representative of normal rainfalls. By contrast, MeteoNet introduced in Larvor & Berthomier (2021) focuses on France and includes precipitation over years. While it has vast samples for moderate precipitation, extreme rainfall events are underrepresented, limiting the evaluation models across the full spectrum of intensities. In this regard, we introduce the KMA dataset, a large-scale collection spanning years from 2014 to 2023 at 10-minute intervals. Owing to the meteorological characteristics of South Korea, it covers a balanced distribution of rainfall intensities from normal to extremely heavy rains. Comparing results on those datasets, we demonstrate that our model generalizes across rainfall distributions and enables more accurate prediction on extreme weather events.

## 3  METHODS

We design our model *exPreCast* motivated by the specific perspective of radar-based extreme precipitation forecasting. First, short-term precipitation is determined by local meteorological features. In addition, extreme rainfall occurs on small region with high intensity, which could be easily smoothed out in image processing. Lastly, forecasting system needs to cover varying time horizons. To address these challenges, we propose a transformer-based architecture, which extends the Video Swin Transformer (Liu et al., 2022) to model spatiotemporal radar sequences. Based on the 3D Swin Transformer, we introduce a domain-specific *Cubic Dual Upsample* (CDU) block in the model to retain fine-grained features and *Temporal Extractor* (TE) block to adjust time horizons. Figure 1 illustrates the overall architecture of the exPreCast. More details are presented in Appendix C.

### 3.1  ENCODER-DECODER 3D SWIN TRANSFORMER

Since rainfall is governed by local meteorological phenomena, we employ a Video Swin Transformer as a backbone to joint spatial and temporal dependencies, focusing on local correlations in windows. The architecture employs an encoder-decoder structure with skip connections to enable multi-scale feature flow. For an input radar volume, the encoder first partitions the data into non-overlapping 3D patches. These patches are then processed through multiple stages of 3D Swin Transformer blocks, with a Patch Merging block at each stage to downsample the features. Within each stage, self-attentions find local correlations between patches in a window, while a shifting mechanism promotes global context and maintains computational efficiency, as introduced in Liu et al. (2022).

The encoded features are further processed by a bottleneck, two 3D Swin Transformer blocks, before being forwarded to the decoder. The decoder mirrors the encoder's structure, using 3D Swin Transformer blocks and CDU blocks to enhance upsampling quality. The CDU blocks fuse interpolation, pixel shuffle, and 3D convolutions. Meanwhile, a spatial-temporal feature passing mechanism, implemented as skip connections, transfers features between the encoder and decoder through either an adding or concatenating operation. Then a Patch Expanding block projects the feature volume to the target spatiotemporal resolution and TE block adjusts the temporal dimension to match the desired forecast lead time, providing flexible prediction horizons.

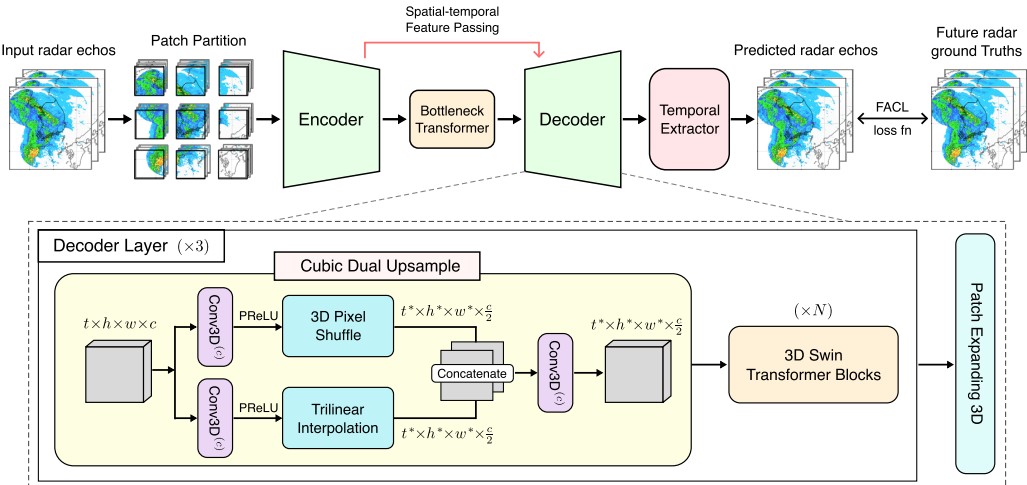

Figure 1: exPreCast Architecture: The encoder compresses input sequences, while the decoder uses the proposed Cubic Dual Upsample (CDU) blocks to reconstruct high-resolution precipitation patterns. The CDU block combines trilinear interpolation and 3D pixel shuffle to refine details and mitigate artifacts. Additionally, a dedicated Temporal Extractor (TE) module enables flexible forecasting across various lead times.

## 3.2 CUBIC DUAL UPSAMPLE BLOCK

Traditional upsampling methods are insufficient for nowcasting extreme events. As shown in Figure 2, standard pixel-shuffle preserves overall structure but loses local patterns and introduces checkerboard artifacts. Conversely, simple interpolation creates aliasing effects, severely degrading image details. To address these limitations and improve the prediction of local precipitation patterns, we introduce the CDU block. Inspired by the dual upsample method from Fan et al. (2022), CDU block fuses trilinear interpolation and 3D pixel-shuffle to improve reconstruction quality and mitigate artifacts in decoded feature volumes. Figure 2 shows that CDU leads to enhanced forecasting performance.

The CDU comprises two parallel branched illustrated in Figure 1: an interpolation branch and a pixel-shuffle branch. While both branch first apply channel-mixing 3D convolution across spatiotemporal axis, convolution in the interpolation branch intends to preserve the channel but one in the other branch expand it to reconstruct high-frequency details. After activation functions and upsamplings in each block, the final 3D convolution fuse the concatenated results.

Given an input feature tensor $z_{in} \in \mathbb{R}^{b \times t \times h \times w \times c}$, the CDU block produces an output $z_{out} \in \mathbb{R}^{b \times t^* \times h^* \times w^* \times \frac{c}{2}}$, where the target resolution is determined by a scale factor $(s_t, s_h, s_w)$ such that $(t^*, h^*, w^*) = (s_t t, s_h h, s_w w)$. Specifically, the trilinear branch produces an intermediate feature $z_{ti} \in \mathbb{R}^{b \times t^* \times h^* \times w^* \times \frac{c}{2}}$ by

$$
\begin{aligned}
z'_{ti} &= \text{Conv3D}^{(c)}(z_{in}) &&\in \mathbb{R}^{b \times t \times h \times w \times c}, \\
z_{ti} &= \text{Conv3D}^{(c)}\left(\text{TI}\left(\sigma\left(z'_{ti}\right)\right)\right) &&\in \mathbb{R}^{b \times t^* \times h^* \times w^* \times \frac{c}{2}},
\end{aligned}
\tag{1}
$$

where TI is trilinear interpolation upsampler, $\sigma$ is PReLU activation and $\text{Conv3D}^{(c)}$ denotes channel-mixing 3D convolution. Similarly, the pixel-shuffle branch outputs $z_{ps} \in \mathbb{R}^{b \times t^* \times h^* \times w^* \times \frac{c}{2}}$:

$$
\begin{aligned}
z'_{ps} &= \text{Conv3D}^{(c)}(z_{in}) &&\in \mathbb{R}^{b \times t \times h \times w \times \frac{1}{2}(s_t s_h s_w c)}, \\
z_{ps} &= \text{Conv3D}^{(c)}\left(\text{PS}\left(\sigma\left(z'_{ps}\right)\right)\right) &&\in \mathbb{R}^{b \times t^* \times h^* \times w^* \times \frac{c}{2}},
\end{aligned}
\tag{2}
$$

where PS is pixel-shuffle upsampler. Then, the final convolution computes an output $z_{out}$ from concatenated those intermediate values:

$$
z_{out} = \text{Conv3D}^{(c)}\left(z_{ti} \oplus z_{ps}\right) \in \mathbb{R}^{b \times t^* \times h^* \times w^* \times \frac{c}{2}},
\tag{3}
$$

where $\oplus$ denotes concatenate operation. This construction effectively alleviates smoothing or artifact issues, yielding coherent and fine-grained precipitation structures. The quantitative comparison on forecasting performance between upsampling blocks is reported in the ablation studies. Further discussion of the effect of the CDU is provided in Appendix D.

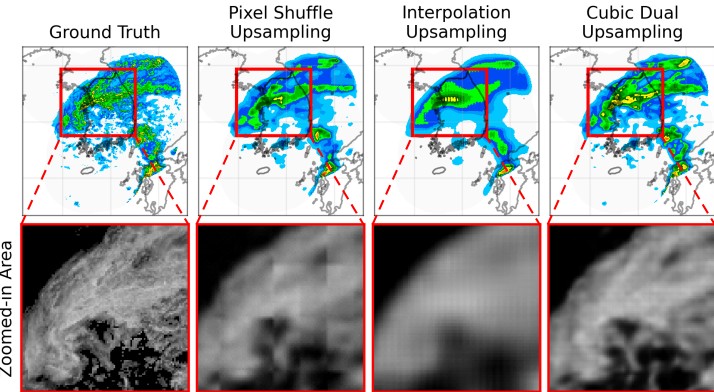

Figure 2: Reconstruction results from different upsampling methods. CDU significantly mitigates checkerboard artifacts and preserves fine-grained radar textures, resulting in better reconstructions.

### 3.3 TEMPORAL EXTRACTOR BLOCK

In precipitation forecasting, the required time horizon varies: very short-term nowcasting is crucial for immediate warnings, while long-term forecasting is more valuable for preparedness. To enable flexible forecasting horizons, we append a TE block after decoder. By configuring the TE block, we can tailor our models to different operational needs. Given the decoder output $Z_{decoder} \in \mathbb{R}^{B \times T \times H \times W \times C}$, the TE block transforms $Z_{decoder}$ of the temporal dimension $T$ into $Y = \text{TE}\,(Z_{decoder})$ of the target forecast horizon $T^*$, using a spatiotemporal 3D convolution:

$$Y = \text{Conv3D}^{(T)}\,(Z_{decoder}) \in \mathbb{R}^{B \times T^* \times H \times W \times C}, \tag{4}$$

where $\text{Conv3D}^{(T)}$ denotes spatiotemporal convolution, which slides over $H, W$ and $C$ dimensions. This allows the model to adapt to varying lead times with minimal computational training cost, facilitating efficient learning across diverse forecasting settings.

Furthermore, the TE block is designed to be bi-lucrative with the decoder framework to maximize performance across different time prediction tasks. For short-term prediction, a small scaling factor in time direction is applied in the CDU decoder part, and the TE block then extracts a minimal but meaningful feature from the decoder, resulting in an efficient and accurate short-term prediction. For long-term prediction, a larger factor is applied along the temporal axis of the CDU decoder. This allows the decoder to learn rich temporal dynamics with the aid of transformer blocks. The TE block then further extracts temporal information, enabling multiple target predictions. Given that both short-term and long-term prediction tasks rely on the same historical information from the past sequences, the encoder part of the model can be shared. This commonality makes it well-suited for a transfer learning approach. We implement this by freezing the encoder's parameters after training on a short-term prediction task. The long-term prediction model is then initialized with this pre-trained encoder, and only the decoder and the TE block are fine-tuned. This provides a highly efficient training framework, reducing the computational cost of developing long-term forecasting models.

### 4 EXPERIMENTS

To rigorously assess the generalization capability of exPreCast, we evaluate it on three distinct datasets: SEVIR, MeteoNet, and KMA. Each dataset reflects a different rainfall distribution, providing complementary perspectives on real-world forecasting scenarios. SEVIR predominantly records extreme rainfall (storm) events, MeteoNet is largely composed of moderate precipitation, while KMA exhibits a more balanced distribution spanning normal to extreme cases. This diversity enables a comprehensive evaluation of the model's robustness across varying rainfall regimes. Detailed descriptions and comparative distributions of these datasets are provided in the Appendix A.

For each dataset, we benchmark exPreCast against several baselines, including state-of-the-art methods, under a fair comparison setting. All baselines are implemented following, as closely as possible, the default configurations reported in their original papers. Our model is trained with the

AdamW optimizer, combined with a warm-up scheduler (Loshchilov & Hutter, 2019), and optimized using the Fourier Amplitude Correlation Loss (Yan et al., 2024). A full description of the training setup and implementation details is provided in the Appendix C.

To assess model performance, we adopt multiple forecasting skill metrics. We primarily use the Critical Success Index (CSI), including CSI-$p$ across thresholds $p$ and averaged CSI-M, evaluated both with and without pooling. CSI with pooling is particularly crucial, as it assesses the model's ability to capture local precipitation patterns–the higher the score, the sharper and more coherent predictions. In addition, following Gao et al. (2022b); Gong et al. (2024), we report the Heidke Skill Score (HSS), which evaluates categorical forecast accuracy relative to random chance. The definitions of these metrics can be found in Appendix B. We also compare efficiency by the number of parameters and inference cost in GFLOPs, providing insights into computational requirements and reproducibility. Furthermore, we provide qualitative comparison with selective baselines in following subsections and Appendix F for the full set of visualizations. Further investigation including adoption of FACL on baselines and other evaluation metrics can be found in Appendices E.1 and E.2, respectively.

## 4.1 KMA

The Korea Meteorological Administration (KMA) officially provides their product to the public via their API hub and briefly explains their product in the document[1]. Among the products from KMA, we use Hybrid Surface Rainfall (HSR) composite radar images, which are generated at a 5-minute interval and cover $1152km \times 1440km$ with a spatial resolution of $500m$. To construct the KMA dataset in this study, we collect all images from 2014 to 2023 with a 10-minute interval, except some timestamps when no data were provided by KMA, and downscale the spatial resolution to $4km$ through uniform subsampling. The downscaling process is clearly illustrated in the Appendix A.

A key distinction of this dataset is its more balanced distribution of events, from normal to extreme, in contrast to the SEVIR and MeteoNet datasets. This characteristic is crucial for evaluating each model's generalization capability. Models are trained to predict the future 6 frames (one hour) from the past 7 frames (one hour). Performance is evaluated after converts predicted radar image via Z-R relation, with thresholds of $[1, 4, 8, 10, 20, 40, 80]$ in mm/h unit.

Table 1: Comparison on KMA with CSI thresholds 20 and 80 across pooling size of 4 and 16

| KMA Model | Parameters(M) | Flops(G) | CSI-M ↑ | | | CSI-20 ↑ | | | CSI-80 ↑ | | | HSS ↑ |
|---|---|---|---|---|---|---|---|---|---|---|---|---|
| | | | POOL1 | POOL4 | POOL16 | POOL1 | POOL4 | POOL16 | POOL1 | POOL4 | POOL16 | |
| UNet | 19.3 | 28 | 0.1822 | 0.1679 | 0.1747 | 0.0520 | 0.0633 | 0.0813 | 0.0016 | 0.0066 | 0.0213 | 0.2680 |
| ConvLSTM | 16.6 | 7 | 0.2419 | 0.2203 | 0.2203 | 0.1234 | 0.1149 | 0.1242 | 0.0140 | 0.0195 | 0.0237 | 0.3532 |
| PhyDNet | 3.09 | 161 | 0.2462 | 0.2197 | 0.2222 | 0.1138 | 0.1066 | 0.1218 | 0.0120 | 0.0160 | 0.0196 | 0.3545 |
| SimVP | 1.9 | 42 | **0.2762** | 0.2558 | 0.2598 | 0.1536 | 0.1444 | 0.1612 | 0.0314 | 0.0381 | 0.0443 | **0.3958** |
| EarthFormer | 15.1 | 61 | 0.2518 | 0.2283 | 0.2302 | 0.1270 | 0.1146 | 0.1256 | 0.0084 | 0.0129 | 0.0165 | 0.3625 |
| AFNO | 61.7 | 63 | 0.1427 | 0.1268 | 0.1082 | 0.0258 | 0.0283 | 0.0287 | 0.0010 | 0.0023 | 0.0083 | 0.2175 |
| CasCast | 391.0 | 1,729 | 0.2573 | 0.3742 | 0.4837 | **0.1702** | 0.3066 | 0.4444 | 0.0346 | 0.0926 | 0.1695 | 0.3806 |
| AlphaPre | 8.5 | 688 | 0.2530 | 0.2311 | 0.2330 | 0.1210 | 0.1154 | 0.1299 | 0.0172 | 0.0204 | 0.0234 | 0.3533 |
| **exPreCast** | 32.0 | 55 | 0.2794 | 0.3721 | 0.4841 | 0.1786 | 0.3013 | 0.4490 | 0.0348 | 0.0835 | 0.1488 | 0.4042 |

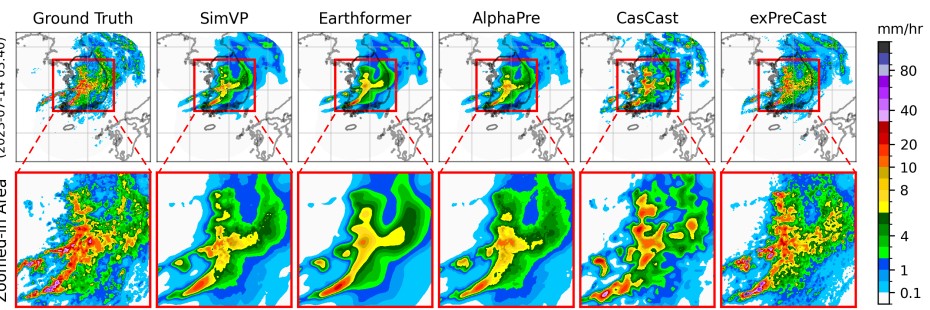

Figure 3: Visualization of 1-hour predictions from different models on the KMA dataset.

Table 1 presents a comparative analysis of forecasting models on the KMA dataset, evaluating their performance via CSI-M, CSI-20, and CSI-80 across multiple pooling levels. Our model demonstrates

[1]Those could be found in *https://apihub.kma.go.kr* and *https://datawiki.kma.go.kr*

superior performance to the majority of baselines and achieves accuracy comparable to the state-of-the-art but computationally intensive CasCast model. A notable finding is that while CasCast slightly outperforms ours in CSI-80 with pooling, exPreCast requires significantly less computational overhead, with over 30 times lower inference GFLOPs and 10 times fewer parameters. On the other hand, our model achieves the highest HSS score on this balanced dataset, which demonstrates its superior categorical forecasting skills and reliable performance under real-world conditions. These results highlight our model's compelling balance of high predictive accuracy and computational efficiency, positioning it as a more practical solution.

To further evaluate prediction quality, we provide a visual comparison of the last prediction output from the top five models ranked by their CSI-M scores presented in Table 1. As shown in Figure 3, SimVP, Earthformer, and AlphaPre exhibit blurry predictions and are unable to identify heavy rainfall events. CasCast, while preserving relatively more detail than the last three models, still fails to precisely detect some heavy rainfall regions. Our model, however, produces the sharpest predictions and accurately identifies the majority of rainfall events.

## 4.2 SEVIR

The Storm Event Imagery (SEVIR) dataset (Veillette et al., 2020) is a commonly used benchmark for evaluating radar-based precipitation forecasting models. The dataset is specifically curated to provide radar observations of extreme rainfall events, capturing storm events over a $384 \times 384$ km region with images at 5-minute intervals for up to 4-hours. Following prior works, we train models to predict the next 12 frames given the past 13 frames. The thresholds for this dataset are based on Vertically Integrated Liquid (VIL), which correspond to pixel values of [16, 74, 133, 160, 181, 219].

Table 2: Comparison on SEVIR with CSI thresholds 181 and 191 across pooling size of 4 and 16

| SEVIR | Parameters(M) | Flops(G) | CSI-M ↑ | | | CSI-181 ↑ | | | CSI-219 ↑ | | | HSS ↑ |
|---|---|---|---|---|---|---|---|---|---|---|---|---|
| Model | | | POOL1 | POOL4 | POOL16 | POOL1 | POOL4 | POOL16 | POOL1 | POOL4 | POOL16 | |
| UNet | 19.3 | 66 | 0.3848 | 0.3830 | 0.3964 | 0.1966 | 0.1926 | 0.2093 | 0.0848 | 0.0853 | 0.0998 | 0.4993 |
| ConvLSTM | 16.6 | 33 | 0.4102 | 0.4163 | 0.4475 | 0.2453 | 0.2525 | 0.2977 | 0.1322 | 0.1380 | 0.1734 | 0.5232 |
| PhyDNet | 13.7 | 701 | 0.4198 | 0.4226 | 0.4410 | 0.2526 | 0.2532 | 0.2782 | 0.1362 | 0.1359 | 0.1526 | 0.5311 |
| SimVP | 2.3 | 132 | 0.4153 | 0.4226 | 0.4530 | 0.2532 | 0.2604 | 0.3000 | 0.1338 | 0.1394 | 0.1685 | 0.5280 |
| EarthFormer | 15.1 | 257 | 0.4310 | 0.4319 | 0.4351 | 0.2622 | 0.2542 | 0.2562 | 0.1448 | 0.1409 | 0.1481 | 0.5411 |
| AFNO | 63.3 | 142 | 0.3613 | 0.3666 | 0.3692 | 0.1558 | 0.1539 | 0.1535 | 0.0859 | 0.0916 | 0.1007 | 0.4706 |
| CasCast | 392.9 | 4,567 | 0.4401 | 0.4640 | 0.5525 | 0.2879 | 0.3179 | 0.3900 | 0.1851 | 0.2127 | 0.2841 | 0.5602 |
| AlphaPre | 32.4 | 5,437 | 0.3996 | 0.4100 | 0.4180 | 0.2173 | 0.2315 | 0.2378 | 0.1081 | 0.1330 | 0.1459 | 0.4996 |
| **exPreCast** | 32.0 | 208 | 0.4179 | **0.4527** | **0.5427** | 0.2568 | **0.2989** | 0.4104 | 0.1468 | **0.1859** | **0.2910** | **0.5430** |

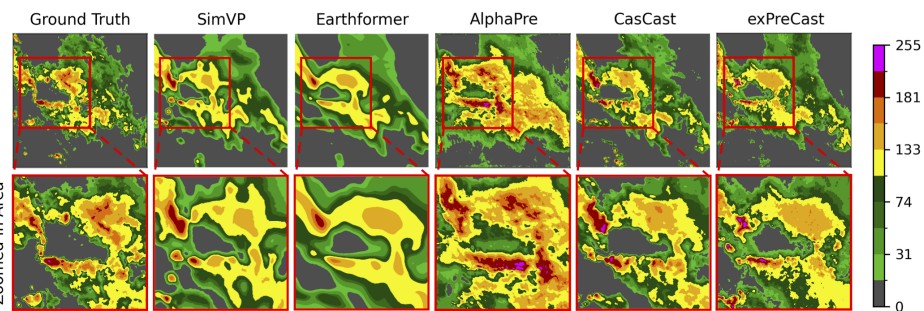

Figure 4: Visualization of 1-hour predictions from different models on the SEVIR dataset.

Table 2 presents the performance comparison on the SEVIR dataset. Our model demonstrates the best overall performance, particularly for CSI-M with pooling 4 and 16, suggesting its effectiveness in preserving precipitation structures. For extreme events (thresholds 181 and 219), exPreCast achieves the highest scores with pooling 4 and 16, a result only exceeded by the CasCast model. On the other hand, our model also shows a high HSS score, demonstrating robust categorical forecasting skills across all precipitation events, even on this dataset curated for extreme rainfall. Notably, the CasCast model requires significantly more computational resources due to its large number of parameters and GFLOPs. In contrast, our model offers a compelling balance of high prediction accuracy and computational efficiency, making it a more practical choice. Figure 4 provides a visual comparison of

the prediction quality on the last frames for different models. Proposed exPreCast once again stands out with its ability to produce sharper predictions and better preserve the structural rainfall patterns.

## 4.3 METEONET

Collected by the French national meteorological service, MeteoNet includes precipitation radar measured every 5 minutes. We use southeastern region and select a top left corner $416 \times 416$ km. Models predict the future 12 frames from the most recent 12 observations. The thresholds evaluated on this dataset are 19, 28, 35, 40, and 47 (dBZ), with 40 and 47 corresponding to the two most extreme rainfall events.

Table 3: Comparison on MeteoNet with CSI thresholds 40 and 47 across pooling size of 4 and 16

| MeteoNet Model | Parameters(M) | Flops(G) | CSI-M ↑ | | | CSI-40 ↑ | | | CSI-47 ↑ | | | HSS ↑ |
|---|---|---|---|---|---|---|---|---|---|---|---|---|
| | | | POOL1 | POOL4 | POOL16 | POOL1 | POOL4 | POOL16 | POOL1 | POOL4 | POOL16 | |
| UNet | 19.3 | 77 | 0.2354 | 0.2168 | 0.1914 | 0.0778 | 0.0708 | 0.0771 | 0.0351 | 0.0382 | 0.0449 | 0.3414 |
| ConvLSTM | 16.6 | 40 | __0.3076__ | 0.2911 | 0.2683 | __0.1642__ | **0.1568** | **0.1646** | __0.1046__ | 0.1030 | 0.1086 | __0.4127__ |
| PhyDNet | 3.09 | 812 | 0.2786 | 0.2556 | 0.2279 | 0.1228 | 0.1114 | 0.1179 | 0.0725 | 0.0682 | 0.0717 | 0.4018 |
| SimVP | 2.3 | 155 | **0.2941** | 0.2785 | 0.2543 | 0.1450 | 0.1387 | 0.1500 | 0.0801 | 0.0806 | 0.0906 | 0.3991 |
| EarthFormer | 15.1 | 309 | 0.2723 | 0.2440 | 0.2155 | 0.1186 | 0.1011 | 0.1026 | 0.0561 | 0.0481 | 0.0472 | 0.3748 |
| AFNO | 63.6 | 166 | 0.2101 | 0.2251 | 0.1818 | 0.0817 | 0.1160 | 0.1125 | 0.0308 | 0.0382 | 0.0372 | 0.3176 |
| AlphaPre | 32.4 | 6,381 | 0.2648 | 0.2421 | 0.2157 | 0.1107 | 0.1126 | 0.1197 | 0.0425 | 0.0430 | 0.0490 | 0.3704 |
| **exPreCast** | 32.0 | 199 | 0.2861 | __0.3452__ | __0.4446__ | **0.1500** | __0.2145__ | __0.3584__ | **0.0856** | __0.1393__ | __0.2525__ | 0.4116 |

Figure 5: Visualization of last-frame predictions from different models on the MeteoNet dataset.

Table 3 summarizes the comparative performance on the MeteoNet dataset. Our exPreCast achieves superior predictive accuracy, particularly with CSI-M at pooling 4 and 16, and demonstrates its capability to preserve fine-grained precipitation patterns even under conditions dominated by normal rainfall events. We have opted to exclude CasCast from this experiment due to its highly unstable results. Even after a fair training setup led to convergence, the CasCast model predictions were consistently noisy and inaccurate (refer to Appendix C). The visual analysis in Figure 5 reveals the superior quality of our model, which not only produces sharper predictions but also effectively preserves the structural integrity of the rainfall patterns.

## 4.4 LONG TIME PREDICTION

Another notable advantage of exPreCast is its flexibility for multi-step-ahead forecasting, which is due to TE block and decoding framework. Beyond the 1-hour (short-term) predictions discussed previously, we also conduct 6-hour (long-term) forecasting experiments on the KMA dataset, which consists of 36 future frames. We achieves this by upsampling the CDU decoder output with a scale factor of 2 along temporal dimension, then using the TE block to extract the targeted 36 frames.

We trained our model through two distinct approaches: training from scratch, and a transfer learning method where we initialized the parameters from a short-term model and then fine-tune only the decoder and TE block while keeping the encoder frozen. We compare our results against other baselines with multi-step-ahead forecasting capabilities, including ConvLSTM, SimVP, and AlphaPre. For a fair comparison, we follow their implementation illustrated in respective paper, adapting the output to 36 frames based on 7 past input frames on the KMA dataset. The results in Table 4 represent that exPreCasts attain the best performance against baseline models, and achieves the highest CSI scores with pooling, demonstrating the ability on local pattern preserving on long-term prediction.

As shown in Figure 6, our model's predictions not only preserve detailed local patterns but are also the only ones capable of capturing heavy rainfall events.

Table 4: Comparison result up to 6 hour predictions on KMA. AlphaPre* is trained using half-scaled dataset due to computational limit. exPreCast[†] presents ours finetuned model.

| KMA-6h Model | Parameters(M) | Flops(G) | CSI-M ↑ | | | CSI-20 ↑ | | | CSI-80 ↑ | | | HSS ↑ |
|---|---|---|---|---|---|---|---|---|---|---|---|---|
| | | | POOL1 | POOL4 | POOL16 | POOL1 | POOL4 | POOL16 | POOL1 | POOL4 | POOL16 | |
| ConvLSTM | 16.6 | 42 | 0.0748 | 0.0623 | 0.0572 | 0.0134 | 0.0137 | 0.0157 | 0.0013 | 0.0021 | 0.0026 | 0.1243 |
| SimVP | 3.5 | 97 | 0.0327 | 0.0260 | 0.0246 | 0.0066 | 0.0065 | 0.0081 | 0.0014 | 0.0023 | 0.0032 | 0.0595 |
| AlphaPre* | 287.2 | 4,898 | 0.0562 | 0.0451 | 0.0464 | 0.0061 | 0.0068 | 0.0095 | 0.0003 | 0.0004 | 0.0006 | 0.0865 |
| **exPreCast** | 33.6 | 229 | 0.0879 | 0.1271 | 0.2262 | 0.0323 | 0.0646 | 0.1542 | 0.0042 | 0.0130 | 0.0415 | 0.1486 |
| **exPreCast[†]** | 25.5 | 229 | **0.1214** | **0.1572** | **0.2289** | **0.0510** | **0.0931** | **0.1708** | **0.0116** | **0.0282** | **0.0642** | **0.1987** |

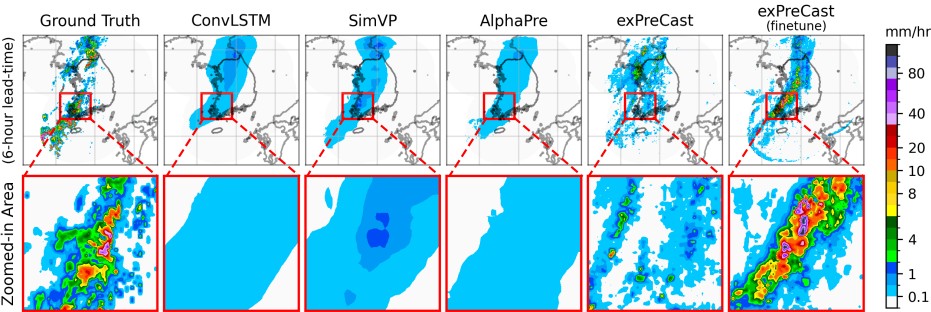

Figure 6: Visualization of 6-hour last-frame predictions from different models on the KMA dataset.

## 4.5 INVESTIGATION ON CDU AND TE

We conduct a targeted ablation study to investigate the performance contribution of the key architectural components: the CDU decoder block and the TE modules.

Our first objective is to rigorously assess the effect of various upsampling strategies. To this end, we compare the proposed CDU block not only with the standard Pixel-Shuffle (PS) and Trilinear Interpolation (TI) methods, but also with other common interpolation techniques, including Bicubic (BIC) and Area (AREA) interpolation, within the decoder framework. Table 5 reports the CSI scores, including those at the final predicted frame, across different thresholds and pooling levels. The results show that CDU achieves the best overall improvement in CSI scores while better preserving local precipitation patterns, as reflected in the pooling metrics. Furthermore, the CSI scores at the final frame reveal that other methods are unable to deliver reliable performance in the long-term prediction, whereas CDU consistently maintains robustness and delivers superior accuracy across all scenarios.

Table 5: Performance comparison of upsampling methods between pixel shuffle (PS), trilinear interpolation (TI) and cubic dual upsample (CDU) across different pooling levels and CSI thresholds, based on 1-hour predictions from the KMA dataset.

| Upsampling Method | CSI-M ↑ | | | CSI-80 ↑ | | | CSI-M (1h) ↑ | | | CSI-80 (1h) ↑ | | |
|---|---|---|---|---|---|---|---|---|---|---|---|---|
| | POOL1 | POOL4 | POOL16 | POOL1 | POOL4 | POOL16 | POOL1 | POOL4 | POOL16 | POOL1 | POOL4 | POOL16 |
| PS | 0.2781 | 0.3577 | 0.4632 | **0.0372** | 0.0797 | 0.1379 | 0.1936 | 0.2577 | 0.3633 | 0.0075 | 0.0287 | 0.0771 |
| TI | 0.2635 | 0.3564 | 0.4740 | 0.0295 | 0.0764 | 0.1436 | 0.1755 | 0.2530 | 0.3884 | 0.0076 | 0.0340 | 0.1023 |
| BIC | 0.2629 | 0.3616 | 0.4814 | 0.0307 | 0.0830 | **0.1530** | 0.1785 | 0.2599 | 0.3937 | 0.0098 | 0.0427 | 0.1156 |
| AREA | 0.2631 | 0.3584 | 0.4765 | 0.0305 | 0.0773 | 0.1466 | 0.1753 | 0.2531 | 0.3879 | 0.0071 | 0.0345 | 0.1053 |
| **CDU** | **0.2794** | **0.3721** | **0.4841** | 0.0348 | **0.0835** | 0.1488 | **0.1963** | **0.2772** | **0.4053** | **0.0116** | **0.0454** | **0.1197** |

On the other hand, to assess the TE block's role in balancing short-term detail preservation and long-term dynamic stability, we conduct experiments by fine-tuning our pretrained model for short-term prediction (1 hour) to various extended lead times, including 2h, 3h, 4h, 5h, and 6h, on KMA dataset. We denote the fine-tuned models using the format "TE-Nf", where N represents the total number of predicted frames (e.g., "TE-12f" indicates a 2-hour prediction model, given a 10-minute frame interval). The CSI scores for these models across the six different prediction horizons are shown in

Table 6. The results demonstrate that by extending the predicted time horizon through the TE block, the model successfully achieves reliable and consistent accuracy across different lead times, without sacrificing the high accuracy achieved on the short-term horizon. This verifies the flexibility and effectiveness of the TE block in adapting the model for various prediction lengths while maintaining forecast quality.

Table 6: Performance comparison of TE modules for different predicted time horizons.

| CSI-M (pool16) | 1h | 2h | 3h | 4h | 5h | 6h |
|---|---|---|---|---|---|---|
| TE-6f | 0.4053 | — | — | — | — | — |
| TE-12f | 0.3770 | 0.2570 | — | — | — | — |
| TE-18f | 0.3854 | 0.2585 | 0.1984 | — | — | — |
| TE-24f | 0.3508 | 0.2302 | 0.1745 | 0.1408 | — | — |
| TE-30f | 0.3840 | 0.2783 | 0.2161 | 0.1752 | 0.1477 | — |
| TE-36f | 0.3923 | 0.2510 | 0.1773 | 0.1347 | 0.1075 | 0.0923 |

## 5 DISCUSSION AND LIMITATIONS

Proving an accurate and efficient precipitation nowcasting model is crucial for human safety, as extreme rainfall events have occurred more frequently over the past decade. However, current AI models often struggle to balance accuracy and efficiency. Existing deterministic models provide insufficient accuracy on extreme events, while generative-model-based ensemble approaches, though highly accurate, require vast computational resources, making them impractical. In this regard, exPreCast would offer a compelling solution by providing high accuracy across diverse rainfall events while maintaining significantly lower computational costs.

The datasets used in this study–KMA, SEVIR, and MeteoNet–differ significantly in scope and characteristics. The KMA dataset is notably extensive, spanning over 10 years and providing a balanced representation of all precipitation scenarios across all seasons. This makes it a highly generalized dataset for evaluating a model's robustness and generalization capability under diverse conditions. In contrast, the SEVIR and MeteoNet datasets are more narrowly focused, primarily concentrating on either intense rainfall or normal events.

While our model offers flexibility across various prediction horizons, achieving strong performance for long-term extreme rainfall events remains challenging. This could stem from both the scarcity of extreme events and fundamentally ill-posed nature of forecasting that relies on local radar observations. We expect that a promising direction is to develop reliable and accurate extrapolation or interpolation to compensate data scarcity, remaining a future work.

Moreover, as a deterministic method, exPreCast produces stable and efficient forecasts but cannot quantify predictive uncertainty. Incorporating uncertainty estimation–potentially through hybrid approaches that integrate probabilistic methods–would further enhance its applicability, particularly in early warning systems.

## 6 CONCLUSION

We introduced exPreCast, a deterministic radar-based precipitation nowcasting model designed to capture fine-grained local rainfall patterns. By combining a cubic dual-upsample decoder with a lightweight temporal extractor, we achieves state-of-the-art accuracy across diverse datasets. These results demonstrate model's superiority and generalizability under different meteorological conditions. Furthermore, exPreCast requires significantly lower computational effort compared to diffusion-based models, implying its potential to serve as a practical complement to real-world forecasting systems.

### ACKNOWLEDGMENTS

This research was supported by the ASTRA Project through the National Research Foundation(NRF) funded by the Ministry of Science and ICT (No. RS-2024-00440063).

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

# A  DATASET DETAILS

## A.1  KMA DATASET

The Korea Meteorological Administration (KMA) dataset provides nationwide radar coverage for South Korea. It consists of various types of radar reflectivity, such as Plan-Position Indicator, Constant-Altitutde PPI, and Hybrid Surface Rainfall (HSR). We used HSR observations obtained every 10 minutes over the central region of South Korea, ranging from longitude $120.5°E$ to $137.5°E$ and latitude $29.0°N$ to $42.0°N$. We normalize each image by clipping negative values to 0 and dividing by 10,000, which results in data values in the range [0, 1) (0.01dBZ). To reduce computational costs while maintaining nationwide coverage of South Korea, we downsampled the data to a 4 km resolution. We then cropped a central $1024 \times 1024$ km square, yielding a $256 \times 256$ input image. The KMA radar dataset spans from 2014 to 2023, with data collected every 10 minutes. For our experiments, we used a time-split approach: data from 2014 to 2021 was used for training, 2022 for validation, and 2023 for testing. The model takes a sequence of 7 frames (one hour of past observations) as input. It is configured to predict either the future 1-hour short-term horizon (6 frames) or the 6-hour long-term horizon (36 frames), with a temporal resolution of 10 minutes per frame.

To evaluate model performance, we first converted the radar reflectivity from dBZ to mm/h using the Marshall-Palmer Z-R relationship, $Z = 200R^{1.6}$, as illustrated in KMA's document `https://datawiki.kma.go.kr`. We then report the CSI-$p$ for thresholds $p = [1, 4, 8, 10, 20, 40, 80]$. Additionally, we applied a mask to exclude pixels outside the radar range, the coverage of which is depicted in Figure 7.

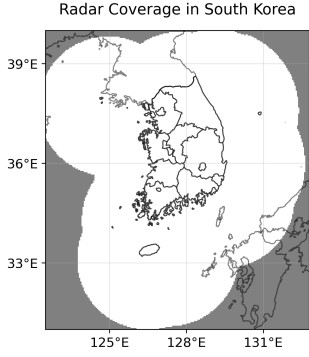

Figure 7: Radar coverage area of South Korea meteorological observatories.

## A.2  SEVIR DATASET

The Storm Event Imagery (SEVIR) dataset (Veillette et al., 2020) is a commonly used benchmark for evaluating radar-based precipitation forecasting models. SEVIR contains meteorological data covering a $384 \times 384$ km region at 5-minute intervals for up to 4-hours, providing satellite imagery, lightning, and radar-based observations such as NEXRAD vertically integrated liquid (VIL). We used the VIL dataset at 1 km resolution and trained a model to predict the next 12 frames given the past 13 frames. In other words, a model takes as input $x \in \mathbb{R}^{13 \times 384 \times 384}$, the most recent 1-hour radar observations, and predicts $y \in \mathbb{R}^{12 \times 384 \times 384}$, corresponding to the next hour. The reported CSI-$p$ scores are based on Vertically Integrated Liquid (VIL), which correspond to pixel values of $p = [16, 74, 133, 160, 181, 219]$. The data value lies in the range $[0, 256)$.

## A.3  METEONET DATASET

MeteoNet is an open meteorological dataset released by Météo-France, the French national meteorological service. It provides a clean and ready-to-use collection of weather data, including radar observations, ground measurements, and satellite imagery. In this work, we use the radar component of the dataset, which spans over three years (January 2016 to October 2018) with a temporal resolution

of 5 minutes. The radar data covers two geographical regions—the north-western and south-eastern quarters of France. We focus on the south-eastern region, where each radar frame has a spatial resolution of $0.01°$ and an image size of $515 \times 784$. For our experiments, we crop the upper-left portion to a size of $416 \times 416$. The forecasting setup uses an input sequence of 12 frames and predicts an output sequence of 12 frames.

The thresholds for categorizing rainfall intensity follow those used in Gong et al. (2024): 0.5, 2, 5, 10, and 30 mm/h. Using the standard Z–R relationship, $R = (Z/200)^{1.6}$, the corresponding reflectivity thresholds are 19, 28, 35, 40, and 47 dBZ. The data lies in the range $[0, 70)$.

### A.4 DATASET DISTRIBUTION AND COMPARISON

Our experiments are conducted on three distinct datasets, each offering a unique perspective on real-world rainfall forecasting. The SEVIR dataset predominantly captures extreme rainfall events, while MeteoNet is primarily composed of moderate precipitation. In contrast, the KMA dataset provides a more balanced distribution, encompassing a full spectrum of cases from normal to extreme. As illustrated in Figure 8, a comparison of the dataset distributions reveals that SEVIR is heavily skewed toward extreme rainfall events and MeteoNet toward moderate precipitation. In contrast, the KMA dataset offers a more balanced distribution, encompassing a broader spectrum of rainfall events. In addition, Table 7 offers a direct quantitative examination for event probability based on KMA's threshold. This characteristic makes the KMA dataset particularly valuable for evaluating a model's generalization capability across diverse conditions.

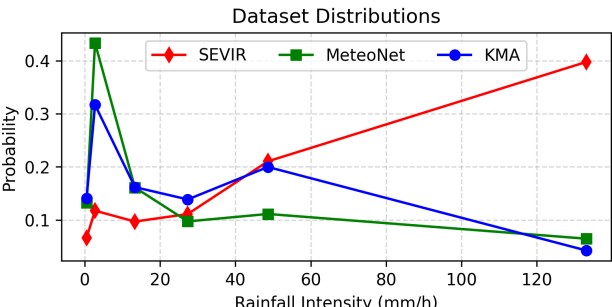

Figure 8: **Distribution comparison of three datasets.** Here, the precipitation unit (mm/h) for the SEVIR dataset is computed following Robinson et al. (2002).

Table 7: Event probability across KMA's threshold.

| Probability | 1 mm/h | 4 mm/h | 8 mm/h | 10 mm/h | 20 mm/h | 40 mm/h | 80 mm/h |
|---|---|---|---|---|---|---|---|
| KMA | 0.18 | 0.15 | 0.05 | 0.16 | 0.17 | 0.17 | 0.13 |
| MeteoNet | 0.22 | 0.21 | 0.09 | 0.14 | 0.14 | 0.09 | 0.11 |

Given the increasing importance of extreme rainfall forecasting, it is crucial to note that extreme heavy rainfall has become more severe over the past decade. As illustrated in Figure 9, a temporal analysis of categorized rainfall events on the KMA and MeteoNet datasets reveals a notable increase in the frequency of shower events. These events are no longer confined to the summer season but are now observed more consistently throughout the year, underscoring the critical need for accurate and robust forecasting models.

## B EVALUATION METRICS

Following Gao et al. (2022b), we compute CSI-$p$ over all predicted frames. Let $\widehat{y}, y \in \mathbb{R}^{T \times H \times W \times C}$ be prediction and ground truth radar reflectivity, where $T$ is the number of output frames, $H, W$ are

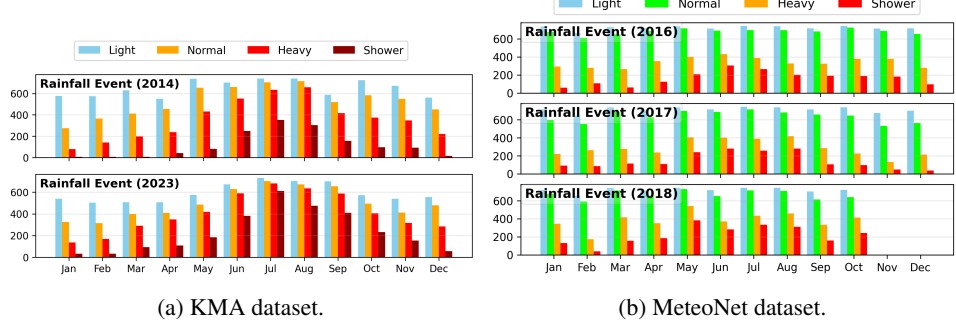

(a) KMA dataset.                                    (b) MeteoNet dataset.

Figure 9: Histograms of rainfall events. Light, Normal, Heavy, and Shower events are categorized by precipitation thresholds: [1, 4, 10, 40] mm/h on KMA and [0.5, 2, 10, 30] mm/h on MeteoNet.

image sizes, and $C$ is the output channels. For each threshold $p$, we define

$$H\left(p, t; \widehat{y}, y\right) = \left|\{h, w, c : \hat{y}\left[t, h, w, c\right] \geq p \text{ and } y\left[t, h, w, c\right] \geq p\}\right|, \tag{5}$$

$$M\left(p, t; \widehat{y}, y\right) = \left|\{h, w, c : \hat{y}\left[t, h, w, c\right] < p \text{ and } y\left[t, h, w, c\right] \geq p\}\right|, \tag{6}$$

$$FA\left(p, t; \widehat{y}, y\right) = \left|\{h, w, c : \hat{y}\left[t, h, w, c\right] \geq p \text{ and } y\left[t, h, w, c\right] < p\}\right|, \tag{7}$$

$$CR\left(p, t; \widehat{y}, y\right) = \left|\{h, w, c : \hat{y}\left[t, h, w, c\right] < p \text{ and } y\left[t, h, w, c\right] < p\}\right|. \tag{8}$$

In other words, $H\left(p, t\right), M\left(p, t\right), FA\left(p, t\right)$ and $CR\left(p, t\right)$ denote the number of pixels correspondent to hits, misses, false alarms, and correct rejections respectively at lead time $t$ and threshold $p$. Given the test set $\{(x_n, y_n)\}_{n=1}^{N}$, these four indicators are counted across all samples and time frames as follows:

$$\overline{H}(p) = \sum_{n=1}^{N} \sum_{t=1}^{T} H\left(p, t; \hat{y}_n, y_n\right), \tag{9}$$

$$\overline{M}(p) = \sum_{n=1}^{N} \sum_{t=1}^{T} M\left(p, t; \hat{y}_n, y_n\right), \tag{10}$$

$$\overline{FA}(p) = \sum_{n=1}^{N} \sum_{t=1}^{T} FA\left(p, t; \hat{y}_n, y_n\right), \tag{11}$$

$$\overline{CR}(p) = \sum_{n=1}^{N} \sum_{t=1}^{T} CR\left(p, t; \hat{y}_n, y_n\right). \tag{12}$$

Then, the CSI-$p$ scores is defined by

$$\text{CSI-}p = \frac{\overline{H}(p)}{\overline{H}(p) + \overline{M}(p) + \overline{FA}(p)}. \tag{13}$$

In addition, the CSI-M score is the average over the thresholds set $P$ and computed by

$$\text{CSI-M} = \frac{1}{|P|} \sum_{p \in P} \text{CSI-}p. \tag{14}$$

CSI is highly sensitive to small spatial misalignment and tends to penalize sharp predictions, overestimating blurred forecast. To compensate this, CSI computed after pooling is suggested in recent studies. For instance, as models produce sharper and more realistic predictions, the gap between CSI with and without pooling increases. In this regard, we measure and report both normal CSI and pooled CSI. Following Ravuri et al. (2021), the pooling of CSI scores measure the forecasting skill on local pattern. It applies max-pooling layer with kernel size of $k$ and stride of $k/4$ on $\widehat{y}, y$ and then computes CSI scores according to the equation 13 and equation 14.

On the other hand, the HSS score, which evaluates categorical forecast accuracy relative to random chance, is computed by

$$\text{HSS} = \frac{1}{|P|} \sum_{p \in P} \frac{2(\overline{H(p)} \times \overline{M(p)} - \overline{FA(p)} \times \overline{CR(p)})}{(\overline{H(p)} + \overline{FA(p)})(\overline{FA(p)} + \overline{M(p)}) + (\overline{H(p)} + \overline{CR(p)})(\overline{CR(p)} + \overline{M(p)})}. \quad (15)$$

## C    EXPERIMENT DETAILS

This section provides a detailed description of our experimental setups. The baseline models were trained using their official code and configurations, which were adapted to each dataset following the specifications in their original papers. Any adjustments to these default settings are explicitly noted here.

### C.1    OURS MODEL TRAINING SETUP

The detailed configuration of our model for 1-hour prediction on KMA is summarized in Table 8. For 6-hour prediction on KMA, detail is listed in Table 9. Training on SEVIR and MeteoNet follows the same configuration with 1-hour KMA. Additionally, the key components are described as follows.

The input radar volume is first partitioned into non-overlapping 3D patches with a patch size of $(M_t, M_h, M_w) = (2, 4, 4)$ along the temporal and spatial dimensions. The encoder consists of three stages with depths $[2, 6, 2]$, meaning each stage includes 2, 6, and 2 3D Swin Transformer blocks, respectively. The decoder mirrors this structure with the same depth configuration $[2, 6, 2]$, and a bottleneck composed of 2 additional 3D Swin Transformer blocks is inserted between the encoder and decoder.

For upsampling, we utilize our proposed Cubic Dual Upsample (CDU) blocks, which apply a scaling factor of either $(1, 2, 2)$ for 1-hour prediction or $(2, 2, 2)$ for 6-hour prediction to upsample the spatial-temporal dimensions. At the final layer of the decoder, a Patch Expanding block is employed using a 3D transposed convolution with kernel size and stride set to $(M_t, M_h, M_w) = (2, 4, 4)$ for 1-hour model or $(1, 4, 4)$ for 6-hour model to recover the full spatiotemporal resolution.

To aid the decoder's learning, we use spatial-temporal feature passing via skip connections, which transfer information from the encoder. The skip connections use either an element-wise adding operation for the 1-hour model or a concatenation operation for the 6-hour model.

Finally, the Temporal Extractor block consists of a 3D convolution, which applies a kernel of size $(3, 3, 1)$ with a stride of $(1, 1, 1)$ and padding of $(1, 1, 0)$ along the $(H, W, C)$ dimensions. The spatial kernel size of $(3, 3)$ enables the model to leverage neighboring spatial context. We note that the TE block extracts 8 temporal features from the decoder's output to predict the target 6 frames for the 1-hour model, and 60 temporal features for the 36 frames of the 6-hour model.

The optimizer we used for model training is the AdamW optimizer with a base learning rate of $1 \times 10^{-3}$. A warm-up cosine learning rate scheduler is applied, with a warm-up ratio of 0.2.

We adopted the FACL loss function to train our model. Denote $p \in [0, 1]$ and let $P(t)$ be a threshold that decreases from 1 to 0 during training. Unlike the stochastic method proposed by Yan et al. (2024), which randomly selects between FAL and FCL based on the condition $p > P(t)$, we observed that this approach leads to unstable training in the early epochs. Consequently, we employ a more stable training strategy by using a linear combination of the two loss components:

$$\text{FACL}(\hat{y}, y, t) = (1 - P(t)) \, \text{FAL} + P(t) \, \text{FCL}. \quad (16)$$

The threshold $P(t)$ decreases linearly from 1 to 0, so

$$P(t) = \frac{1 - n}{N}, \quad (17)$$

where $N$ is the total number of training iterations.

For training on the SEVIR dataset, we set the total number of epochs to 100, consistent with the training setup of other baseline models. Training on other datasets, the maximum number of iterations is set to 100,000 with a batch size of 16. Model performance is validated every 5,000 iterations using

the CSI-M score at the last frame prediction, and the model with the highest score is selected for final evaluation.

We are equipped with four NVIDIA A6000 GPUs ($4 \times 48$ GB) to train our model. Nevertheless, training on 1-hour KMA requires only one A6000 GPU ($\approx 28$GB); training on SEVIR, MeteoNet, and 6-hour KMA require full four A6000 GPUs.

## C.2 BASELINE MODELS

The baseline models in this paper—UNet (Veillette et al., 2020), ConvLSTM (Shi et al., 2015), PhyDNet (Guen & Thome, 2020), SimVP (Gao et al., 2022a), Earthformer (Gao et al., 2022b), AFNO (Pathak et al., 2022), CasCast (Gong et al., 2024), and AlphaPre (Lin et al., 2025)—were configured following their respective papers, with the exception of CasCast and AlphaPre due to the computational limitation. The CasCast model, which is a diffusion-based approach, leverages deterministic backbones such as SimVP or Earthformer to produce a 10-member ensemble for high-resolution nowcasting. Due to its substantial inference cost (exceeding one hundred thousand GFLOPs), reproducing its full ensemble-based approach was computationally infeasible. To allow for a fair comparison, we adapted the CasCast model to generate a single prediction. In the case of the AlphaPre model, its heavy reliance on Fourier Neural Operator structures in Amplitude Network results in significant memory consumption. To relief the computational barrier, we reduced the spatial dimensions $(H, W)$ of the input data by a factor of 2 for longer prediction on the KMA.

## C.3 TRAINING LOSSES OF CASCAST

Figure 10 represents the training losses of CasCast. The training process consists of 3 steps: training deterministic model (earthformer), training autoencoder, and training diffusion based on the latent data. Although diffusion model shows lower loss in MeteoNet, it might be overfitted to ambiguous latent data given by unstable autoencoder and deterministic model. This may cause weird reconstruction shown in Appendix F.

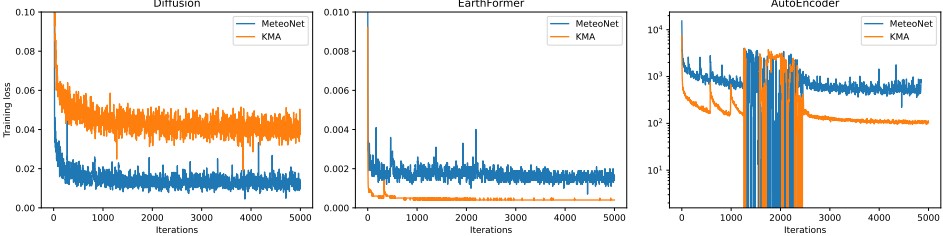

Figure 10: Training loss of CasCast on MeteoNet and KMa datasets.

# D EMPIRICAL ANALYSIS ON CDU

An empirical analysis is conducted as follows to provide an intuitive understanding of how the proposed CDU successfully improves the upsampling performance and mitigates the unrealistic artifacts generated by the Pixel Shuffle or Trilinear Interpolation methods.

The analysis assumes that the target image $Y$ can be decomposed into a smooth low-frequency component $Y_{low}$ and a detailed high-frequency (residual) component $Y_{high}$. That is,

$$Y = Y_{low} + Y_{high}$$

First, traditional interpolation methods (e.g., Trilinear Interpolation) generate overly smooth results, where the underlying target image contains highly nonlinear behavior between pixels. Therefore, the TI branch primarily approximates only the low-frequency content of the target image:

$$TI(X) \approx Y_{low}$$

Table 8: The details of the exPreCast model on 1-hour KMA dataset. `(S)WMSA` refers to the (Shifted) Window Multi-Head Self-Attention mechanism, following the design introduced in Liu et al. (2022). In this configuration, non-shifted attention is applied in odd-numbered Swin Transformer blocks, while the shifted window mechanism is used in even-numbered blocks.

| Block | Layer | Resolution | Channels |
|---|---|---|---|
| Input | - | $7 \times 256 \times 256$ | 1 |
| Patch Partition & Linear Embedding | Conv3D | $7 \times 256 \times 256 \to 4 \times 128 \times 128$ | $1 \to 96$ |
| 3D Swin Transformer Blocks $\times 2$ | LayerNorm | $4 \times 128 \times 128$ | 96 |
| | (S)WMSA | $4 \times 128 \times 128$ | 96 |
| | LayerNorm | $4 \times 128 \times 128$ | 96 |
| | MLP | $4 \times 128 \times 128$ | 96 |
| Patch Merging | Linear | $4 \times 128 \times 128 \to 4 \times 64 \times 64$ | $96 \to 192$ |
| | LayerNorm | $4 \times 64 \times 64$ | 192 |
| 3D Swin Transformer Blocks $\times 6$ | LayerNorm | $4 \times 64 \times 64$ | 192 |
| | (S)WMSA | $4 \times 64 \times 64$ | 192 |
| | LayerNorm | $4 \times 64 \times 64$ | 192 |
| | MLP | $4 \times 64 \times 64$ | 192 |
| Patch Merging | Linear | $4 \times 64 \times 64 \to 4 \times 32 \times 32$ | $192 \to 384$ |
| | LayerNorm | $4 \times 32 \times 32$ | 384 |
| 3D Swin Transformer Blocks $\times 2$ | LayerNorm | $4 \times 32 \times 32$ | 384 |
| | (S)WMSA | $4 \times 32 \times 32$ | 384 |
| | LayerNorm | $4 \times 32 \times 32$ | 384 |
| | MLP | $4 \times 32 \times 32$ | 384 |
| Patch Merging | Linear | $4 \times 32 \times 32 \to 4 \times 16 \times 16$ | $384 \to 768$ |
| | LayerNorm | $4 \times 16 \times 16$ | 768 |
| 3D Swin Transformer Blocks $\times 2$ | LayerNorm | $4 \times 16 \times 16$ | 768 |
| | (S)WMSA | $4 \times 16 \times 16$ | 768 |
| | LayerNorm | $4 \times 16 \times 16$ | 768 |
| | MLP | $4 \times 16 \times 16$ | 768 |
| Cubic Dual Upsample | Conv3D | $4 \times 16 \times 16$ | $768 \to 1536$ |
| | PReLU | $4 \times 16 \times 16$ | 1536 |
| | PixelShuffle3D | $4 \times 16 \times 16 \to 4 \times 32 \times 32$ | $1536 \to 384$ |
| | Conv3D | $4 \times 32 \times 32$ | 384 |
| | Conv3D | $4 \times 16 \times 16$ | 768 |
| | PReLU | $4 \times 16 \times 16$ | 768 |
| | Upsampling | $4 \times 16 \times 16 \to 4 \times 32 \times 32$ | 768 |
| | Conv3D | $4 \times 32 \times 32$ | $768 \to 384$ |
| | Concatenate | $4 \times 32 \times 32$ | $368 \to 768$ |
| | Conv3D | $4 \times 32 \times 32$ | $768 \to 384$ |
| | LayerNorm | $4 \times 32 \times 32$ | 384 |
| Feature Passing | SkipConnection (Adding) | $4 \times 32 \times 32$ | 384 |
| 3D Swin Transformer Blocks $\times 2$ | LayerNorm | $4 \times 32 \times 32$ | 384 |
| | (S)WMSA | $4 \times 32 \times 32$ | 384 |
| | LayerNorm | $4 \times 32 \times 32$ | 384 |
| | MLP | $4 \times 32 \times 32$ | 384 |
| Cubic Dual Upsample | Conv3D | $4 \times 32 \times 32$ | $384 \to 768$ |
| | PReLU | $4 \times 32 \times 32$ | 768 |
| | PixelShuffle3D | $4 \times 32 \times 32 \to 4 \times 64 \times 64$ | $768 \to 192$ |
| | Conv3D | $4 \times 64 \times 64$ | 192 |
| | Conv3D | $4 \times 32 \times 32$ | 384 |
| | PReLU | $4 \times 32 \times 32$ | 384 |
| | Upsampling | $4 \times 32 \times 32 \to 4 \times 64 \times 64$ | 384 |
| | Conv3D | $4 \times 64 \times 64$ | $384 \to 192$ |
| | Concatenate | $4 \times 64 \times 64$ | $192 \to 384$ |
| | Conv3D | $4 \times 64 \times 64$ | $384 \to 192$ |
| | LayerNorm | $4 \times 64 \times 64$ | 192 |
| Feature Passing | SkipConnection (Adding) | $4 \times 64 \times 64$ | 192 |
| 3D Swin Transformer Blocks $\times 6$ | LayerNorm | $4 \times 64 \times 64$ | 192 |
| | (S)WMSA | $4 \times 64 \times 64$ | 192 |
| | LayerNorm | $4 \times 64 \times 64$ | 192 |
| | MLP | $4 \times 64 \times 64$ | 192 |
| Cubic Dual Upsample | Conv3D | $4 \times 64 \times 64$ | $192 \to 384$ |
| | PReLU | $4 \times 64 \times 64$ | 384 |
| | PixelShuffle3D | $4 \times 64 \times 64 \to 4 \times 128 \times 128$ | $384 \to 96$ |
| | Conv3D | $4 \times 128 \times 128$ | 96 |
| | Conv3D | $4 \times 64 \times 64$ | 192 |
| | PReLU | $4 \times 64 \times 64$ | 192 |
| | Upsampling | $4 \times 64 \times 64 \to 4 \times 128 \times 128$ | 192 |
| | Conv3D | $4 \times 128 \times 128$ | $192 \to 96$ |
| | Concatenate | $4 \times 128 \times 128$ | $96 \to 192$ |
| | Conv3D | $4 \times 128 \times 128$ | $192 \to 96$ |
| | LayerNorm | $4 \times 128 \times 128$ | 96 |
| Feature Passing | SkipConnection (Adding) | $4 \times 128 \times 128$ | 96 |
| 3D Swin Transformer Blocks $\times 2$ | LayerNorm | $4 \times 128 \times 128$ | 96 |
| | (S)WMSA | $4 \times 128 \times 128$ | 96 |
| | LayerNorm | $4 \times 128 \times 128$ | 96 |
| | MLP | $4 \times 128 \times 128$ | 96 |
| Patch Expanding | TransposeConv3D | $4 \times 128 \times 128 \to 8 \times 256 \times 256$ | $96 \to 1$ |
| Temporal Extractor | Conv3D | $8 \times 256 \times 256 \to 6 \times 256 \times 256$ | 1 |

Table 9: The details of the exPreCast model on 6-hour KMA dataset. (S)WMSA refers to the (Shifted) Window Multi-Head Self-Attention mechanism, following the design introduced in Liu et al. (2022). In this configuration, non-shifted attention is applied in odd-numbered Swin Transformer blocks, while the shifted window mechanism is used in even-numbered blocks.

| Block | Layer | Resolution | Channels |
|---|---|---|---|
| Input | - | $7 \times 256 \times 256$ | 1 |
| Patch Partition & Linear Embedding | Conv3D | $7 \times 256 \times 256 \to 4 \times 128 \times 128$ | $1 \to 96$ |
| 3D Swin Transformer Blocks $\times 2$ | LayerNorm | $4 \times 128 \times 128$ | 96 |
| | (S)WMSA | $4 \times 128 \times 128$ | 96 |
| | LayerNorm | $4 \times 128 \times 128$ | 96 |
| | MLP | $4 \times 128 \times 128$ | 96 |
| Patch Merging | Linear | $4 \times 128 \times 128 \to 4 \times 64 \times 64$ | $96 \to 192$ |
| | LayerNorm | $4 \times 64 \times 64$ | 192 |
| 3D Swin Transformer Blocks $\times 6$ | LayerNorm | $4 \times 64 \times 64$ | 192 |
| | (S)WMSA | $4 \times 64 \times 64$ | 192 |
| | LayerNorm | $4 \times 64 \times 64$ | 192 |
| | MLP | $4 \times 64 \times 64$ | 192 |
| Patch Merging | Linear | $4 \times 64 \times 64 \to 4 \times 32 \times 32$ | $192 \to 384$ |
| | LayerNorm | $4 \times 32 \times 32$ | 384 |
| 3D Swin Transformer Blocks $\times 2$ | LayerNorm | $4 \times 32 \times 32$ | 384 |
| | (S)WMSA | $4 \times 32 \times 32$ | 384 |
| | LayerNorm | $4 \times 32 \times 32$ | 384 |
| | MLP | $4 \times 32 \times 32$ | 384 |
| Patch Merging | Linear | $4 \times 32 \times 32 \to 4 \times 16 \times 16$ | $384 \to 768$ |
| | LayerNorm | $4 \times 16 \times 16$ | 768 |
| 3D Swin Transformer Blocks $\times 2$ | LayerNorm | $4 \times 16 \times 16$ | 768 |
| | (S)WMSA | $4 \times 16 \times 16$ | 768 |
| | LayerNorm | $4 \times 16 \times 16$ | 768 |
| | MLP | $4 \times 16 \times 16$ | 768 |
| Cubic Dual Upsample | Conv3D | $4 \times 16 \times 16$ | $768 \to 3072$ |
| | PReLU | $4 \times 16 \times 16$ | 3072 |
| | PixelShuffle3D | $4 \times 16 \times 16 \to 8 \times 32 \times 32$ | $3072 \to 384$ |
| | Conv3D | $8 \times 32 \times 32$ | 384 |
| | Conv3D | $4 \times 16 \times 16$ | 768 |
| | PReLU | $4 \times 16 \times 16$ | 768 |
| | Upsampling | $4 \times 16 \times 16 \to 8 \times 32 \times 32$ | 768 |
| | Conv3D | $4 \times 32 \times 32$ | $768 \to 384$ |
| | Concatenate | $8 \times 32 \times 32$ | $368 \to 768$ |
| | Conv3D | $8 \times 32 \times 32$ | $768 \to 384$ |
| | LayerNorm | $8 \times 32 \times 32$ | 384 |
| Feature Passing | Concatenate | $(8+4) \times 32 \times 32$ | 384 |
| 3D Swin Transformer Blocks $\times 2$ | LayerNorm | $12 \times 32 \times 32$ | 384 |
| | (S)WMSA | $12 \times 32 \times 32$ | 384 |
| | LayerNorm | $12 \times 32 \times 32$ | 384 |
| | MLP | $12 \times 32 \times 32$ | 384 |
| Cubic Dual Upsample | Conv3D | $12 \times 32 \times 32$ | $384 \to 1536$ |
| | PReLU | $12 \times 32 \times 32$ | 1536 |
| | PixelShuffle3D | $12 \times 32 \times 32 \to 24 \times 64 \times 64$ | $1536 \to 192$ |
| | Conv3D | $24 \times 64 \times 64$ | 192 |
| | Conv3D | $12 \times 32 \times 32$ | 384 |
| | PReLU | $12 \times 32 \times 32$ | 384 |
| | Upsampling | $12 \times 32 \times 32 \to 24 \times 64 \times 64$ | 384 |
| | Conv3D | $24 \times 64 \times 64$ | $384 \to 192$ |
| | Concatenate | $24 \times 64 \times 64$ | $192 \to 384$ |
| | Conv3D | $24 \times 64 \times 64$ | $384 \to 192$ |
| | LayerNorm | $24 \times 64 \times 64$ | 192 |
| Feature Passing | Concatenate | $(24+4) \times 64 \times 64$ | 192 |
| 3D Swin Transformer Blocks $\times 6$ | LayerNorm | $28 \times 64 \times 64$ | 192 |
| | (S)WMSA | $28 \times 64 \times 64$ | 192 |
| | LayerNorm | $28 \times 64 \times 64$ | 192 |
| | MLP | $28 \times 64 \times 64$ | 192 |
| Cubic Dual Upsample | Conv3D | $28 \times 64 \times 64$ | $192 \to 768$ |
| | PReLU | $28 \times 64 \times 64$ | 768 |
| | PixelShuffle3D | $28 \times 64 \times 64 \to 56 \times 128 \times 128$ | $768 \to 96$ |
| | Conv3D | $56 \times 128 \times 128$ | 96 |
| | Conv3D | $28 \times 64 \times 64$ | 192 |
| | PReLU | $28 \times 64 \times 64$ | 192 |
| | Upsampling | $28 \times 64 \times 64 \to 56 \times 128 \times 128$ | 192 |
| | Conv3D | $56 \times 128 \times 128$ | $192 \to 96$ |
| | Concatenate | $56 \times 128 \times 128$ | $96 \to 192$ |
| | Conv3D | $56 \times 128 \times 128$ | $192 \to 96$ |
| | LayerNorm | $56 \times 128 \times 128$ | 96 |
| Feature Passing | Concatenate | $(56+4) \times 128 \times 128$ | 96 |
| 3D Swin Transformer Blocks $\times 2$ | LayerNorm | $60 \times 128 \times 128$ | 96 |
| | (S)WMSA | $60 \times 128 \times 128$ | 96 |
| | LayerNorm | $60 \times 128 \times 128$ | 96 |
| | MLP | $60 \times 128 \times 128$ | 96 |
| Patch Expanding | TransposeConv3D | $60 \times 128 \times 128 \to 60 \times 256 \times 256$ | $96 \to 1$ |
| Temporal Extractor | Conv3D | $60 \times 256 \times 256 \to 36 \times 256 \times 256$ | 1 |

On the other hand, while the Pixel Shuffle (PS) can capture the high-frequency part, it typically attempts to reconstruct the entire image in one step:

$$PS(X) \approx Y = Y_{low} + Y_{high}$$

However, it is empirically observed that forcing PS to learn both low- and high-frequency details simultaneously often leads to undesirable checkerboard artifacts in the output (as demonstrated in Figure 2). Consequently, our proposed CDU, which fuses the TI and PS branches, leverages residual learning. By allowing the TI branch to capture the dominant $Y_{low}$, the task of the PS branch is effectively decoupled. This enables the PS branch to focus primarily on reconstructing the high-frequency residual $Y_{high}$. That is,

$$PS(X) \rightarrow (Y - TI(X)) \approx Y_{high}$$

Therefore, by decoupling the reconstruction task, where the TI branch handles the smooth low-frequency content and the PS branch efficiently targets the high-frequency residuals, the CDU successfully mitigates the unrealistic artifacts while ensuring high structural fidelity in the final prediction.

## E  MORE QUANTITATIVE RESULTS

### E.1  IMPACT OF FACL ON BASELINE MODELS

The performance superiority of our exPreCast model results from the synergistic integration of the proposed CDU decoder, TE blocks, and the utilized FACL loss function. While FACL contributes to an improvement in predictive texture and forecasting quality, its effective integration within our architecture yields the best overall results. To validate the coupling effect and demonstrate the intrinsic strength of exPreCast's design, we conducted experiments where the default loss functions of strong baselines were replaced with FACL. This comparison includes the top three baselines, ranked by their CSI-M in Table 1. The results, detailed in Table 10, show that although FACL indeed provides a general performance boost to these existing models, exPreCast still achieves the highest overall performance. This outcome confirms that the superior performance is primarily attributable to the specific architectural innovations of exPreCast.

Table 10: Impact of the FACL loss function on baseline models.

| Model | Loss | CSI-M ↑ | | | CSI-20 ↑ | | | CSI-80 ↑ | | | HSS ↑ |
|---|---|---|---|---|---|---|---|---|---|---|---|
| | | POOL1 | POOL4 | POOL16 | POOL1 | POOL4 | POOL16 | POOL1 | POOL4 | POOL16 | |
| SimVP | - | 0.2762 | 0.2558 | 0.2598 | 0.1536 | 0.1444 | 0.1612 | 0.0314 | 0.0381 | 0.0443 | 0.3958 |
| | FACL | 0.2691 | 0.3548 | 0.4717 | 0.1722 | 0.2890 | 0.4388 | 0.0282 | 0.0710 | 0.1305 | 0.3912 |
| Earthformer | - | 0.2518 | 0.2283 | 0.2302 | 0.1270 | 0.1146 | 0.1256 | 0.0084 | 0.0129 | 0.0165 | 0.3625 |
| | FACL | 0.1597 | 0.2712 | 0.3762 | 0.0519 | 0.1605 | 0.3121 | 0.0063 | 0.0332 | 0.0948 | 0.2417 |
| AlphaPre | - | 0.2530 | 0.2311 | 0.2330 | 0.1210 | 0.1154 | 0.1299 | 0.0172 | 0.0204 | 0.0234 | 0.3533 |
| | FACL | 0.2539 | 0.3334 | 0.4367 | 0.1540 | 0.2549 | 0.3844 | 0.0232 | 0.0586 | 0.1000 | 0.3663 |
| **exPreCast** | FACL | **0.2794** | **0.3721** | **0.4841** | **0.1786** | **0.3013** | **0.4490** | **0.0348** | **0.0835** | **0.1488** | **0.4042** |

### E.2  OTHER EVALUATION METRICS

Throughout this paper, the CSI scores serve as the primary performance indicator, aligning with established meteorological literature due to its robust field standardization and domain-expert interpretability (e.g., ECMWF Technical Memoranda No. 430, see Nurmi (1994)). Nevertheless, to ensure a comprehensive validation, we also evaluate performance using several other common metrics frequently utilized in nowcasting literature: Mean Absolute Error (MAE), Mean Square Error (MSE), Fractional Skill Scores (FSS), and Regional Histogram Divergence (RHD) (proposed in Yan et al. (2024)). While MAE/MSE assess point-wise accuracy commonly used in image prediction, FSS and RHD offer valuable insights into distribution preservation and structural fidelity. These metrics were evaluated across all trained baselines and our model on the KMA dataset, with FSS/RHD using a neighbor size of 15 and consistent precipitation thresholds. The full comparative results are detailed in Table 11.

Table 11: Performance evaluation using various metrics on the KMA dataset.

| Model | Parameters(M) | Flops(G) | MAE ↓ | MSE ↓ | FSS ↑ | RHD ↓ |
|---|---|---|---|---|---|---|
| UNet | 19.3 | 28 | 0.0753 | 10.6032 | 0.4802 | 0.0478 |
| ConvLSTM | 16.6 | 7 | 0.0524 | 0.5755 | 0.5419 | 0.0366 |
| PhyDNet | 3.09 | 161 | **0.0488** | 0.5474 | 0.5264 | 0.0377 |
| SimVP | 1.9 | 42 | **0.0482** | **0.5219** | 0.5594 | 0.0325 |
| Earthformer | 15.1 | 61 | 0.0514 | 0.5579 | 0.5542 | 0.0352 |
| AFNO | 61.7 | 63 | 0.0622 | 0.7400 | 0.4332 | 0.0632 |
| CasCast | 391.0 | 1,729 | 0.0958 | 3.1821 | **0.6108** | **0.0178** |
| AlphaPre | 8.5 | 688 | 0.0491 | **0.5420** | 0.5720 | 0.0363 |
| **exPreCast (Ours)** | 32.0 | 55 | 0.0639 | 3.1342 | **0.5950** | **0.0122** |

The results in Table 11 further validate exPreCast's superiority, demonstrating its ability to not only achieve high skill (CSI) but also to produce outputs with high structural fidelity (FSS) and accurate intensity distributions (RHD). Although our model achieves comparable accuracy to most baselines on point-wise metrics (MAE/MSE), it is crucial to note the inherent limitations: point-wise metrics are generally considered insufficient for robust precipitation forecasting evaluation. As widely recognized, MSE tends to smooth out high-intensity regions, leading to performance underestimation in extreme events. Therefore, CSI remains the most effective metric for capturing performance across various event intensity levels. Furthermore, we emphasize exPreCast's efficiency advantage: while a competitor like CasCast may show a marginally better FSS score, its computational overhead (GFLOPs) is more than 30 times greater than ours, underscoring the superior utility of our proposed model.

# F    MORE QUALITATIVE RESULTS

## F.1    VARIOUS MODELS PREDICTIONS ON KMA DATASET

We present more qualitative results from various model predictions on KMA dataset, as shown in Figures 11, 12 and 13.

## F.2    VARIOUS MODELS PREDICTIONS ON SEVIR DATASET

We present more qualitative results from various model predictions on SEVIR dataset, as shown in Figures 14, 15, and 16.

## F.3    VARIOUS MODELS PREDICTIONS ON METEONET DATASET

We present more qualitative results from various model predictions on MeteoNet dataset, as shown in Figures 17, 18, and 19.

## F.4    SIX HOURS PREDICTIONS ON KMA DATASET

We present here the qualitative results from 6-hour predictions models studied in Section 4.4. Figure 20 visualizes the prediction every hour.

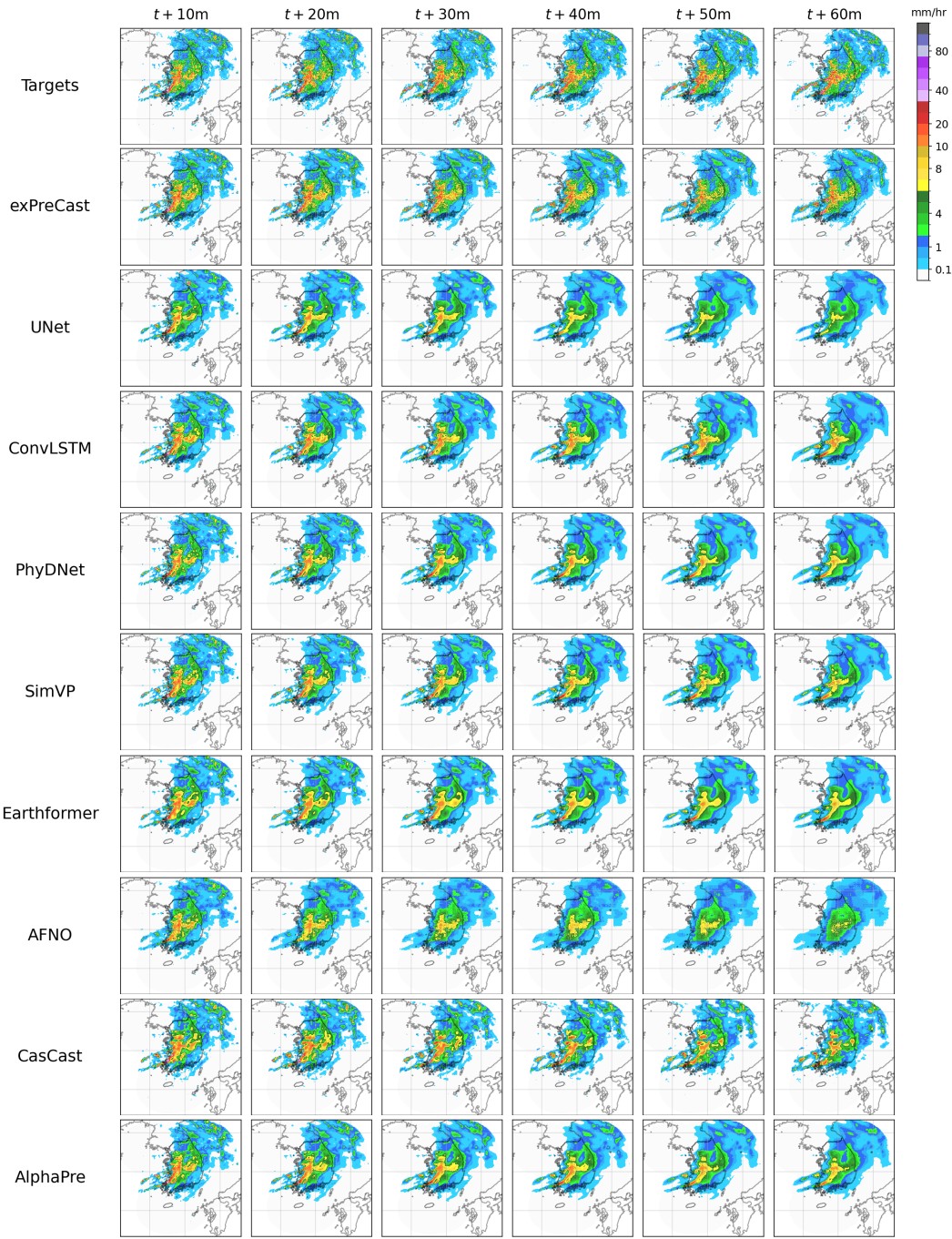

Figure 11: A set of example forecasts on KMA.

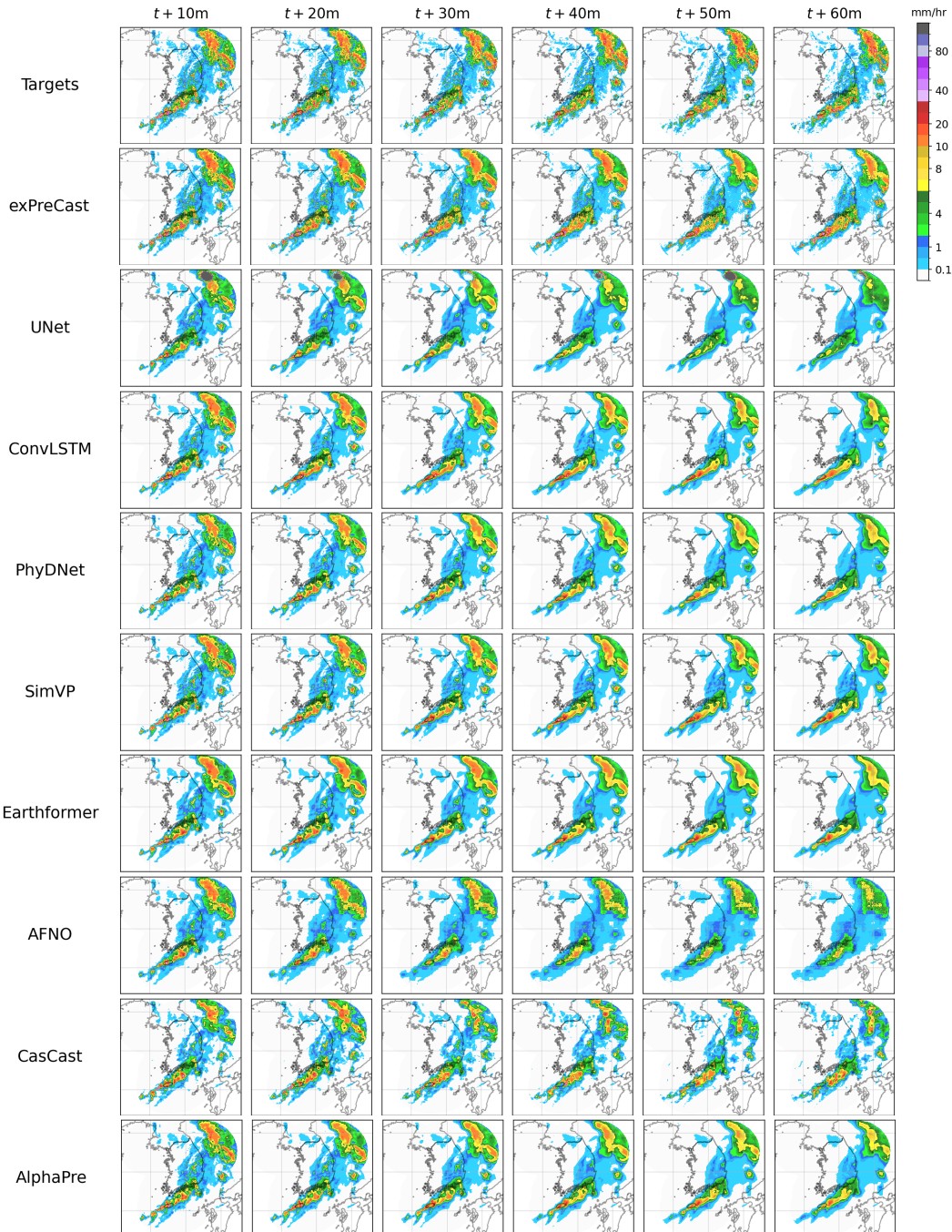

Figure 12: A set of example forecasts on KMA.

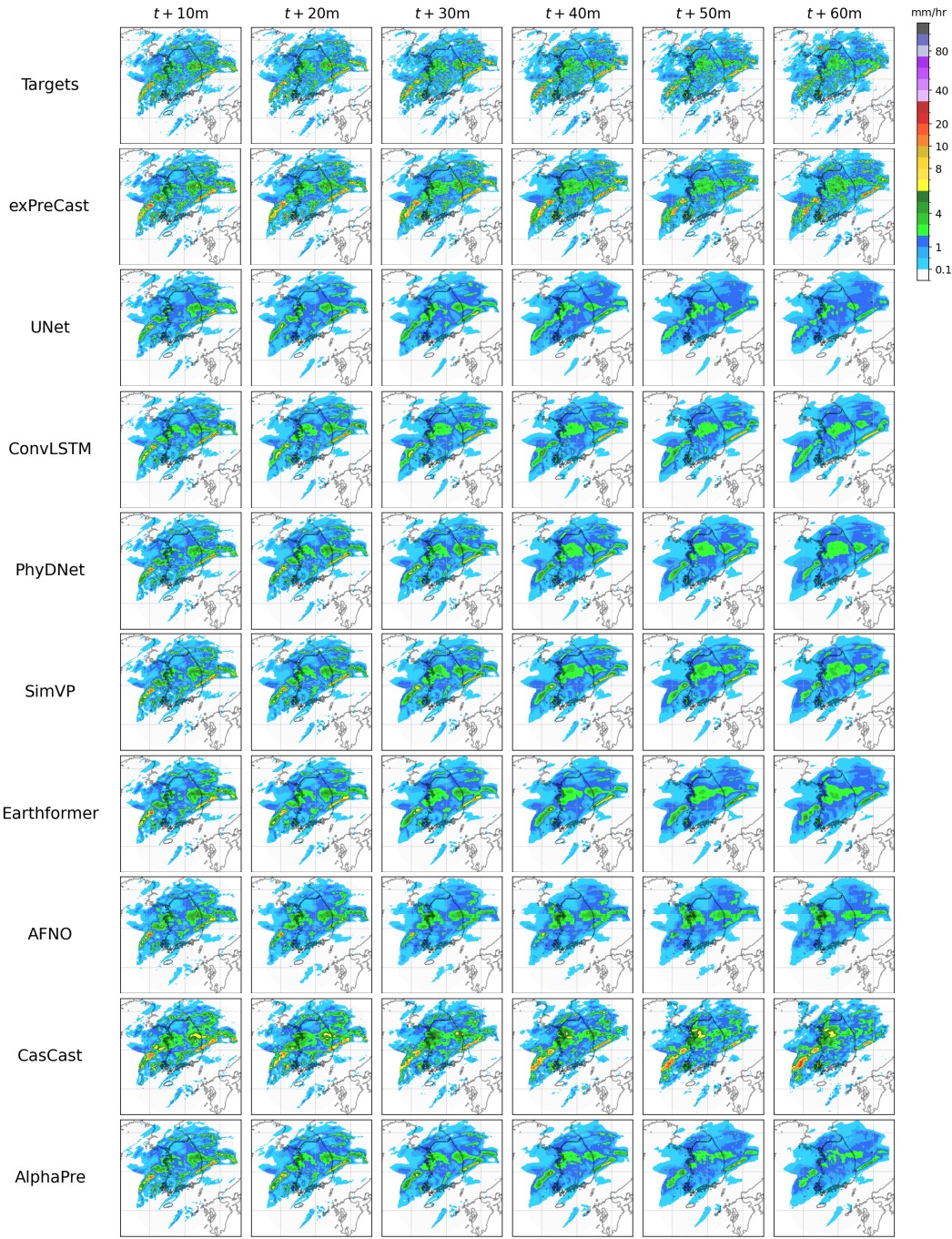

Figure 13: A set of example forecasts on KMA.

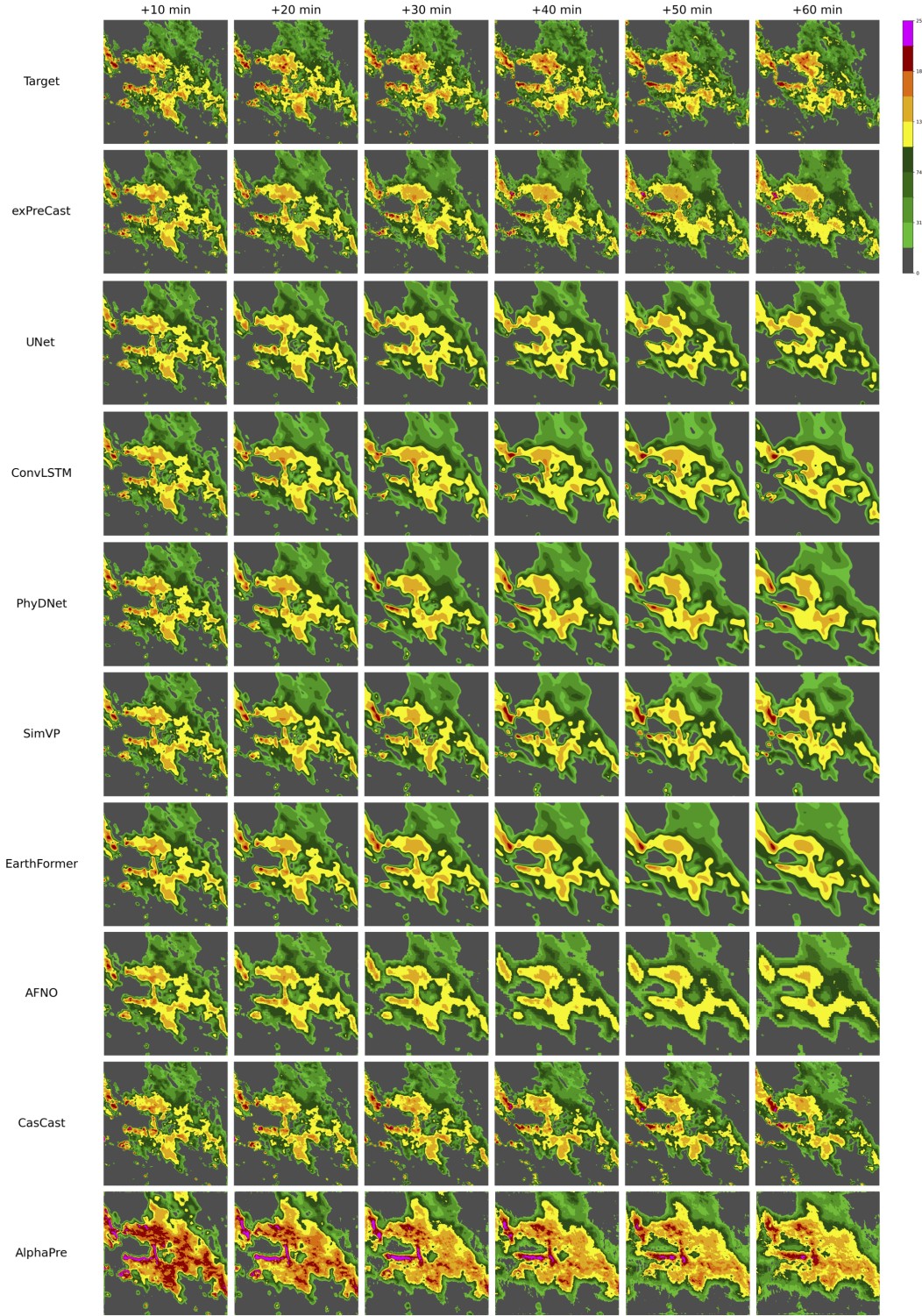

Figure 14: A set of example forecasts on SEVIR.

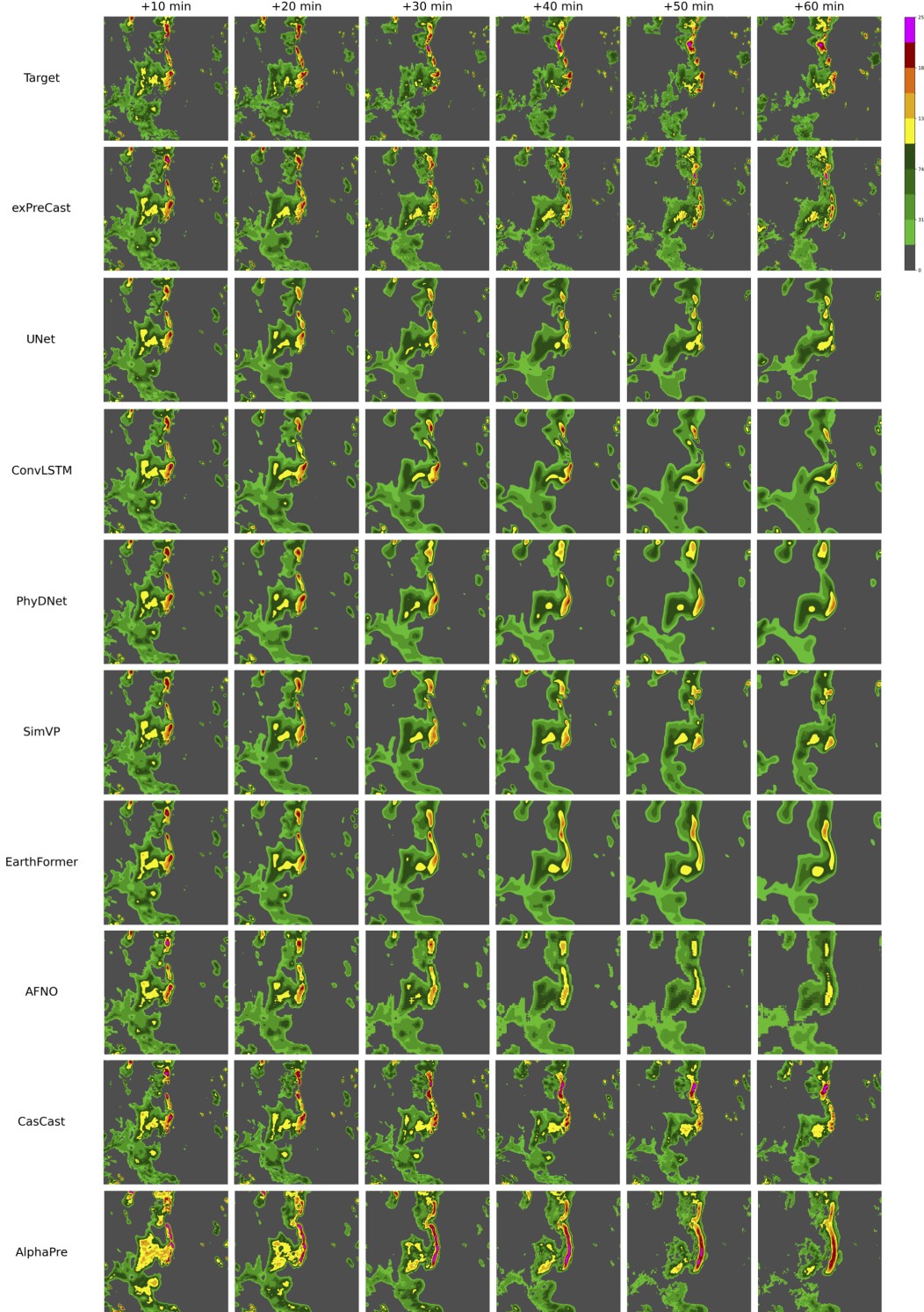

Figure 15: A set of example forecasts on SEVIR.

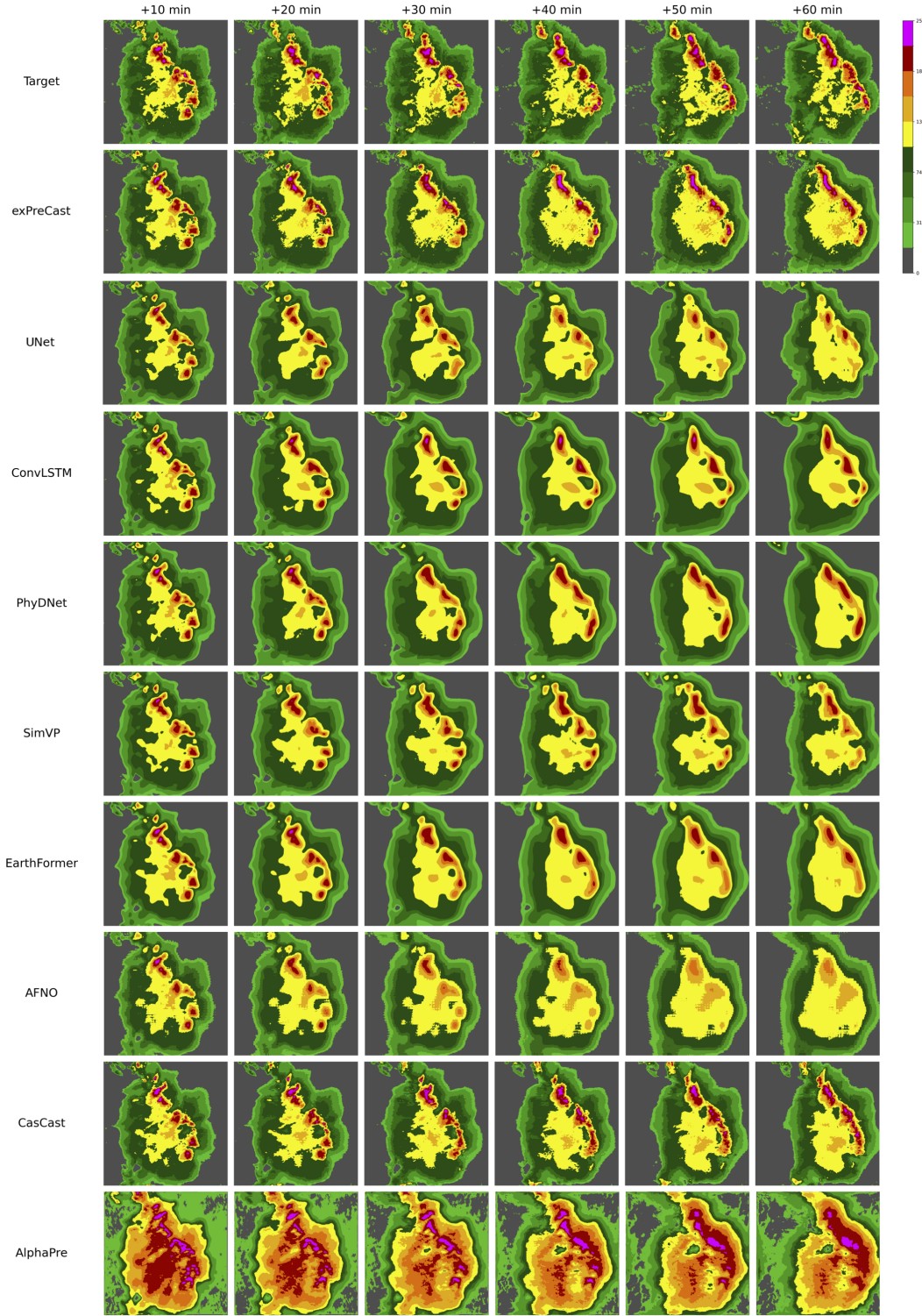

Figure 16: A set of example forecasts on SEVIR.

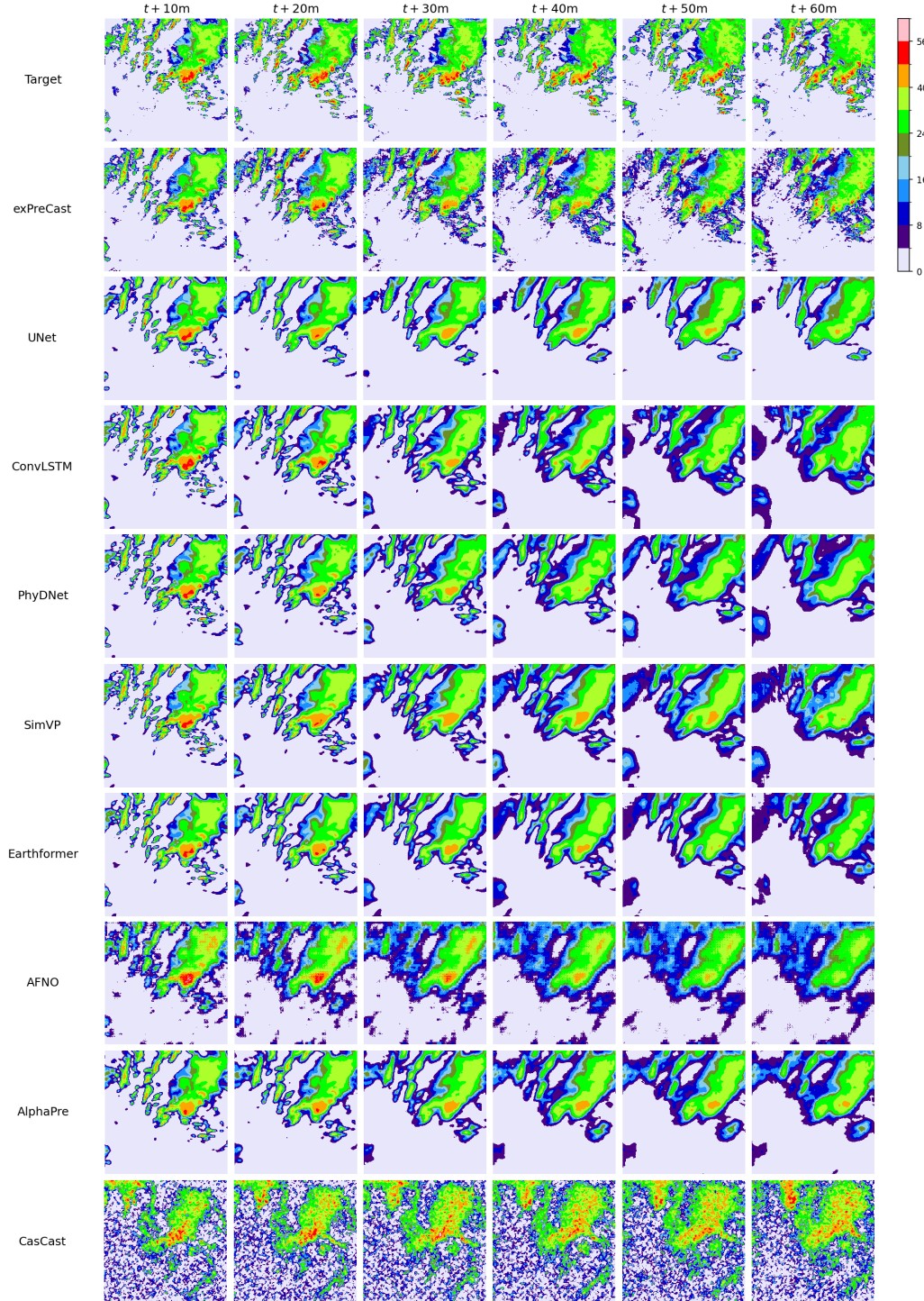

Figure 17: A set of example forecasts on MeteoNet.

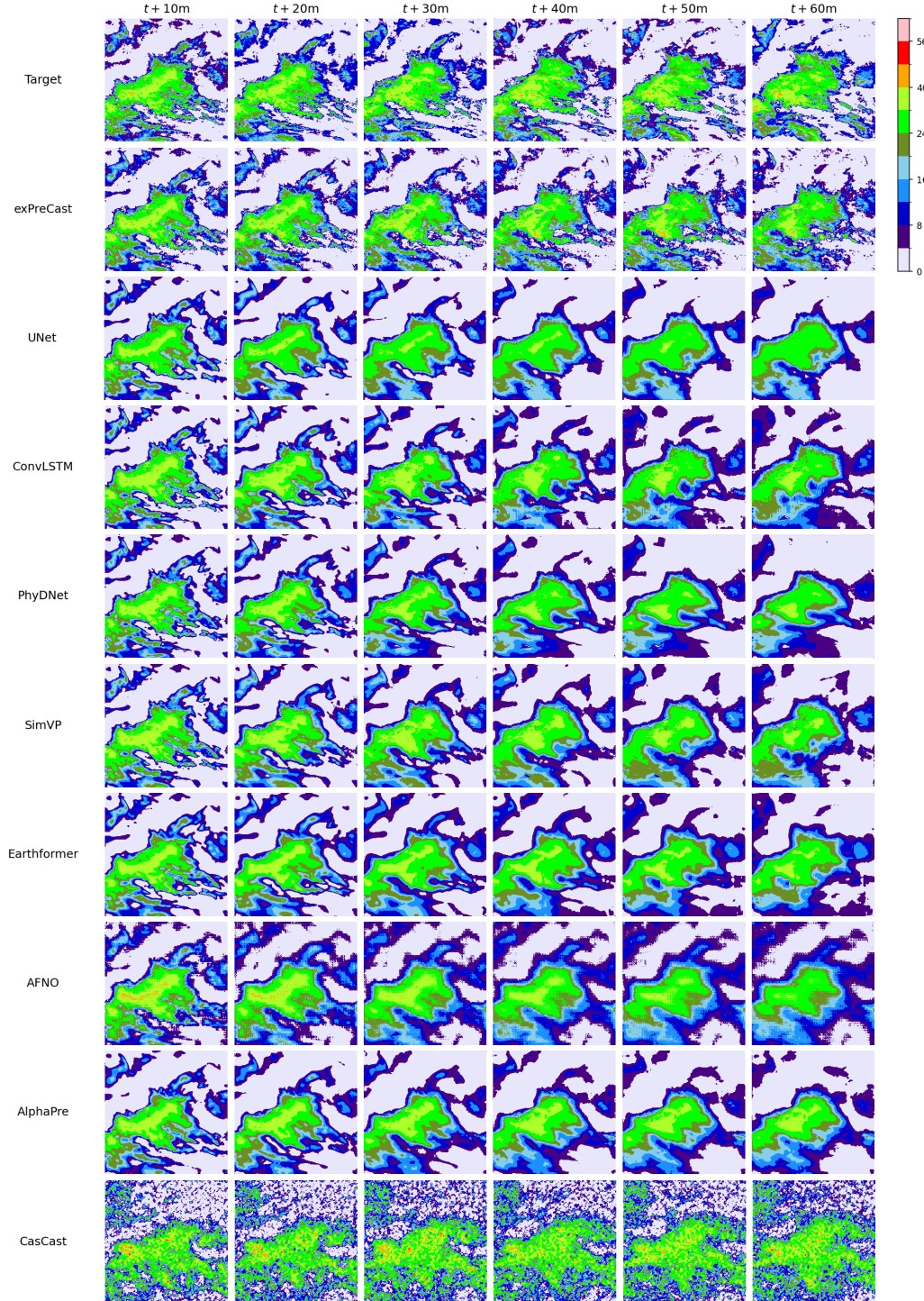

Figure 18: A set of example forecasts on MeteoNet.

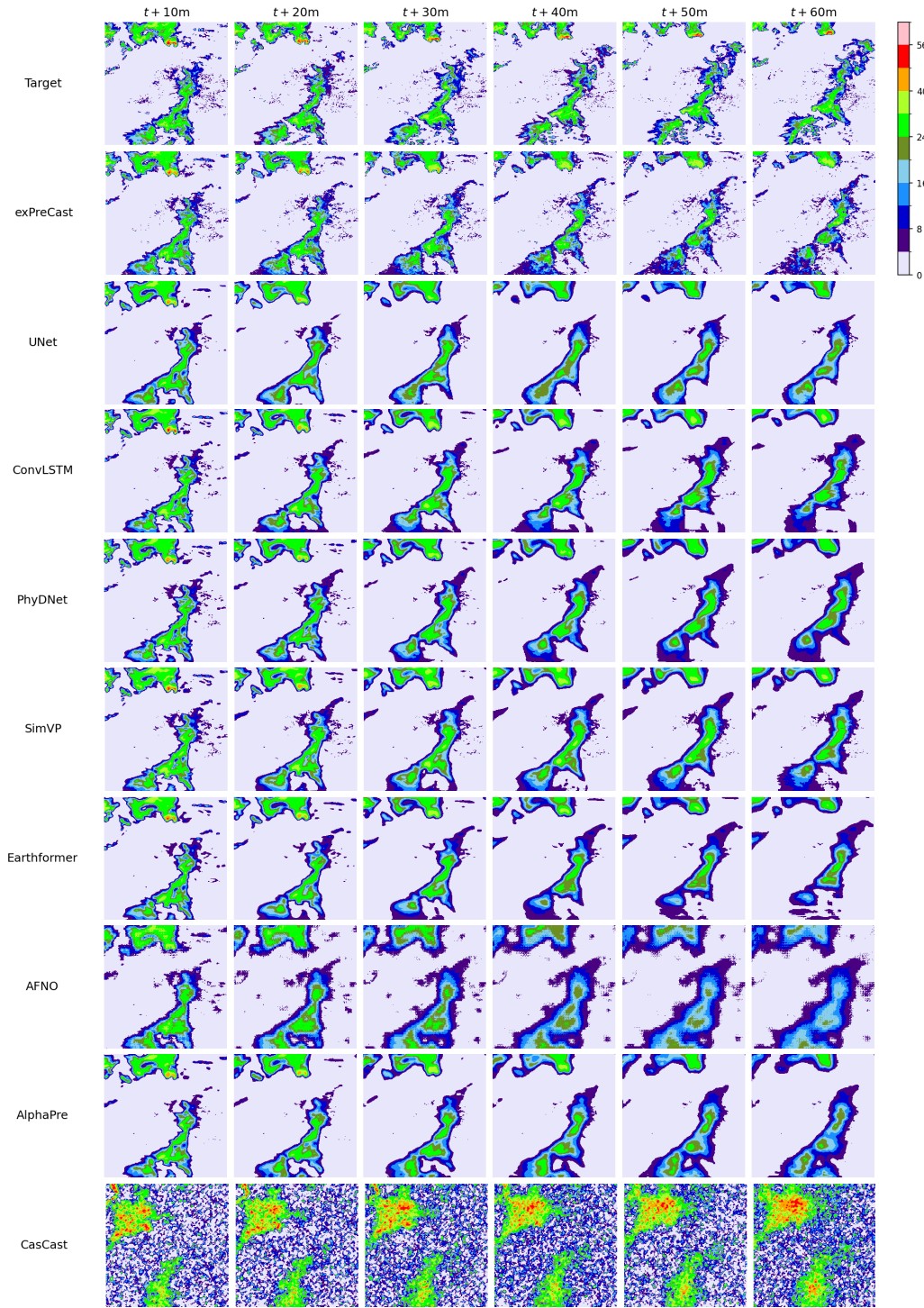

Figure 19: A set of example forecasts on MeteoNet.

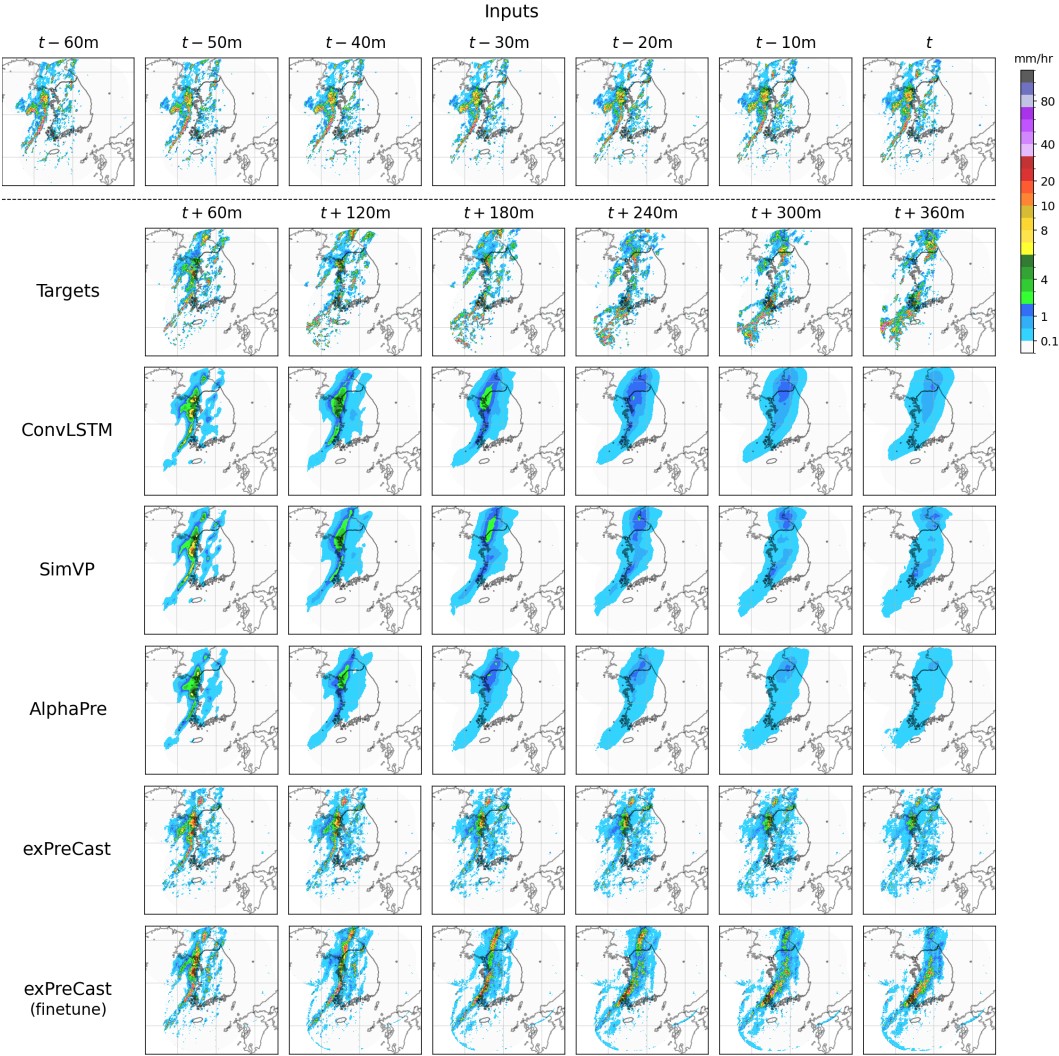

Figure 20: A set of example forecasts on 6-hour KMA.

## G    BROADER IMPACT

This work focuses on high-resolution precipitation nowcasting, which plays a critical role in anticipating localized extreme rainfall events. By enabling more accurate short-term forecasts at fine spatial scales, the proposed model has the potential to significantly improve public safety and disaster preparedness. In particular, it can support early warning systems that help protect lives and reduce property damage during extreme weather events. Accurate and timely rainfall prediction is especially crucial in densely populated regions, where even short delays in response can lead to severe consequences. While the model demonstrates good performance, careful coordination with operational forecasting systems and transparent communication with stakeholders are essential to ensure its effective and responsible deployment.

