# OpenReview forum: "Extreme Weather Nowcasting via Local Precipitation Pattern Prediction"
_ICLR.cc/2026/Conference — ICLR 2026 Poster_

### Official Review · Reviewer_aEjv · 2025-10-24

**Soundness:** 3
**Presentation:** 2
**Contribution:** 2
**Rating:** 6
**Confidence:** 3

**Summary:**

This paper proposes exPreCast, an efficient deterministic deep learning framework for short-term precipitation (nowcasting), focusing on extreme rainfall events. exPreCast is built upon the Video Swin Transformer and introduces a Cubic Dual Upsample (CDU) decoder to preserve fine radar texture and a Temporal Extractor (TE) for flexible forecasting horizons and dynamic adjustment. The authors also construct a balanced and extensive radar dataset (from the Korea Meteorological Administration, KMA) to complement existing imbalanced benchmarks like SEVIR and MeteoNet. Experiments across all three datasets show that exPreCast achieves state-of-the-art or comparable performance with far less computational cost, making it feasible for real-time applications. Ablation studies further support the efficacy of the proposed modules, especially the CDU. The paper also discusses limitations and potential future directions.

**Strengths:**

The paper cleverly integrates efficient CDU and TE modules, significantly improving precipitation nowcasting accuracy and generalizability for both extreme and normal regimes, while maintaining low computational cost—making it highly practical for real-world use.

**Weaknesses:**

While there is an ablation comparison of upsampling methods, there is no systematic ablation of other core modules—such as the backbone transformer, TE module, or skip connection. The contributions of each component if removed or replaced are not separately quantified, making it hard to attribute performance gains.

Although the KMA dataset is introduced as intermediately balanced, details about its collection, labeling standards, preprocessing, and physical data quality are thin. There is also little about public release or ensuring its fairness and community usability.

**Questions:**

Could the authors present ablation results for the TE, backbone transformer, skip connections etc., and analyze how removing each affects performance and failure patterns?

Can the authors clarify the details of KMA data collection, labeling, preprocessing, and release? What measures ensure its broad representativeness and high physical credibility?

---

> ### Author Response · Authors · 2025-11-21
>
> **W1. ... ablation of other core modules--such as ...**
> We sincerely appreciate the reviewer's insightful suggestions regarding the completeness of the ablation study. We agree that quantifying the unique contribution of each core component is essential for validating the claimed performance gains. To comprehensively address this, we have conducted extensive ablation studies on the remaining core modules. These experiments examine the influence of the backbone design, the feature passing mechanism (or skip-connection), and the TE block across various forecast horizons.
> The results of these analyses are presented in the tables below:
> - Ablation Study on Core Component (Backbone, Skip-connection, and Upsampling).
> This ablation investigates the individual impact of the architecture's fundamental elements. Here, CDU* applied to 2D Swin is just dual upsample block, which is proposed in Fan et al. This case aims to study the importance of local attention in temporal direction.
>
> | | Components | | CSI-M| | | CSI-80| | | CSI-M(1h) | | | CSI-80(1h) | | |
> |--|--|--| --|--|--| --|--|--| --|--|--| --|--|--|
> | Backbone | Upsampling | Feature-passing | pool1 | pool4 | pool16 | pool1 | pool4 | pool16 |pool1 | pool4 | pool16 |pool1 | pool4 | pool16 |
> |2D Swin| CDU* | O | 0.1616 | 0.2713 | 0.3869 | 0.0056 | 0.0254 | 0.0718 | 0.1574 | 0.2500 | 0.3321 | 0.0066 | 0.0302 | 0.0828 |
> | 3D Swin | CDU | X | 0.1713 | 0.2685 | 0.3417 | 0.0055 | 0.0276 | 0.0831 | 0.1564 | 0.2269 | 0.2710 | 0.0041 | 0.0199 | 0.0608 |
> |3D Swin| PS | O | 0.2781 | 0.3577 | 0.4632 | **0.0372** | 0.0797 | 0.1379 | 0.1936 | 0.2577 | 0.3633 | 0.0075 | 0.0287 | 0.0771 |
> |3D Swin | TRI | O | 0.2635 | 0.3564 | 0.4740 | 0.0295 | 0.0764 | 0.1436 | 0.1755 | 0.2530 | 0.3884 | 0.0076 | 0.0340 | 0.1023 |
> |3D Swin | CDU | O | **0.2794** | **0.3721** | **0.4841** | 0.0348 | **0.0835** | **0.1488** | **0.1963** | **0.2772** | **0.4053** | **0.0116** | **0.0454** | **0.1197** |
>
> - Ablation Study on Temporal Extractor (TE) Block.
> This ablation studies the TE block's specific role in balancing short-term detail preservation and long-term dynamic stability by comparing the model fine-tuned for different time horizons:
> | CSI-M (pool16) |   |      |      |      |     |  |
> |--|--|--|--|--|--|--|
> |  | 1h | 2h | 3h | 4h | 5h | 6h |
> |TE-6f| 0.4053 | -- | -- | -- | -- | -- |
> |TE-12f | 0.3770 | 0.2570 | -- | -- | -- | -- |
> |TE-18f | 0.3854 | 0.2585 | 0.1984 | -- | -- | -- |
> |TE-24f | 0.3508 | 0.2302 | 0.1745 | 0.1408 | -- | -- |
> |TE-30f | 0.3840 | 0.2783 | 0.2161 | 0.1752 | 0.1477 | -- |
> |TE-36f | 0.3923 | 0.2510 | 0.1773 | 0.1347 | 0.1075 | 0.0923 |
>
> Here, TE-6f stands for the 6 frames extracted from the TE block. The results demonstrate the effectiveness and flexibility of the TE block. By extending the predicted time horizon via the TE block, the model maintains reliable and consistent accuracy across various lead times, without compromising the high fidelity achieved in short-term forecasts. This study indicates that the TE block efficiently adapts the model to different prediction lengths while sustaining overall forecast quality.

---

> ### Author Response · Authors · 2025-11-21
>
> **W2. ... its (KMA's) collection, labeling standards, preprocessing, ...**
> We thank the reviewer for pointing that the construction of the KMA dataset needs more details. First, the Korea Meteorological Administration (KMA) officially released the composite radar products to the public. KMA provides these products through its API hub (*https://apihub.kma.go.kr*) and documents their radar processing, quality-control procedures, and product specifications in its official data portal (*https://datawiki.kma.go.kr*). The internal pipeline from raw radar observations to the final composite fields is fully operated and quality-controlled by KMA, not by us. We will clarify the links in the revised version.
> For this study, we use Hybrid Surface Rainfall (HSR) composite images from 2014 to 2023. The products are provided at 5-minute intervals and cover $1152km\times1440km$ at $500m$ resolution, consisting of $2305\times 2881$ pixels. We construct our dataset by reducing its size according to the following steps. First, we download all available images at a 10-minute interval. Some timestamps are missing solely because the KMA service did not provide data at those times; we do not filter any samples by ourselves. Following the Lambert conformal conic projection described in KMA’s document, we compute the area range from latitude $29.0^\circ$ to $42.0^\circ$ and longitude $120.5^\circ$ to $137.5^\circ$. Then, we crop the central $1024km\times1024km$ of the area and uniformly subsample with stride 8 to generate $256\times256$ images with the spatial resolution of $4km$. Lastly, we vertically flip to ensure north-south ordering aligned with increasing latitude. As KMA affirmed that the constructed dataset could be open to the public when properly attributed, we will share those dataset as well as the source codes.
> Since there was no manual labeling or quality-control procedures in these processes, we rely on the KMA for the credibility. Nevertheless, as KMA serves as South Korea’s official governmental meteorological authority and participates in the World Meteorological Organization (WMO), the composite radar products it generates undergo internationally aligned quality-control standards, and thus can be regarded as highly credible.

---

> > ### Comment · Reviewer_aEjv · 2025-11-23
> >
> > Thank you for the authors' response. I believe a score of 6 is appropriate for this paper, and I will therefore maintain my rating.

---

### Official Review · Reviewer_Fimn · 2025-10-31

**Soundness:** 3
**Presentation:** 4
**Contribution:** 3
**Rating:** 6
**Confidence:** 4

**Summary:**

The proposed model exPreCast is a Transformer-based deterministic precipitation nowcasting model that integrates local spatiotemporal attention, texture-preserving CDU decoder and TE block to flexibly adjust forecast lead times, which also constructs a balanced KMA dataset containing ordinary and extreme precipitation. Experiments on KMA, SEVIR and MeteoNet show exPreCast’s  outperform baselines like ConvLSTM and SimVP.

**Strengths:**

1. ExPreCast constructs a balanced KMA dataset containing both ordinary and extreme precipitation, addressing the imbalance of existing datasets, and providing more comprehensive data support for evaluating model generalization.
2. ExPreCast integrates the CDU decoder and TE block, enabling the model to perform well in both 1-hour short-term and 6-hour long-term forecasts.

**Weaknesses:**

1. This work lacks a comparison with GAN-based methods, and adding such a comparison can provide a more complete assessment of exPreCast’s performance.
2. This work would be better to include verification related to the impact of the TE block in the ablation study section to quantify the effect of the TE block on balancing the detail preservation of short-term forecasts and the dynamic stability of long-term forecasts.
3. Relying on CSI as an indicator may not fully judge the quality of precipitation predictions. The average intensity of precipitation predictions will significantly affect CSI. The authors can consider adding the scores which aims to assess how closely the nowcasting outputs.

**Questions:**

The questions here are related to the three points described as weaknesses.

---

> ### Author Response · Authors · 2025-11-21
>
> **W1.  ... comparison with GAN-based methods ...**
> We thank the reviewer for the valuable suggestion on the baselines. Following the comment, we included an additional comparison against STIP, a recent GAN-based video prediction model that has demonstrated competitive performance in high-resolution spatiotemporal forecasting. We re-implemented STIP on the KMA dataset under the same evaluation protocol. As shown in the table, exPreCast consistently outperforms STIP, especially under heavy-rain cases, highlighting our model’s capability to preserve fine-scale precipitation structures even under extreme events.
>
> | | |CSI-M| | | CSI-20| | | CSI-80| |
> |--|--|--|--|--|--|--|--|--|--|
> |Model| pool1| pool4 | pool16 | pool1| pool4 | pool16 | pool1| pool4 | pool16 |
> |STIP| 0.1814 | 0.2428 | 0.3430 | 0.0845 | 0.1707 | 0.2981 | 0.0124 | 0.0476 | 0.1078 |
> |exPreCast| 0.2794 | 0.3721 | 0.4841 | 0.1786 | 0.3013 | 0.4490 | 0.0348 | 0.0835 | 0.1488 |
>
> Indeed, we would like to note that in recent years diffusion-based generative models(such as CasCast) have surpassed GAN-based ones in both fidelity and forecasting skill for spatiotemporal predictions. In this regard, it is not surprising that diffusion-based approaches outperform GAN-based models in weather forecasting. Although we were able to run STIP on the KMA dataset due to time constraints, we will include those results as well if additional experiments on SEVIR and MeteoNet are completed in the remaining periods.
>
> **W2.  ... impact of the TE block in the ablation study section ...**
> We appreciate the reviewer for suggesting this insightful ablation study regarding the impact of the TE block. Quantifying the TE block's role in balancing short-term detail preservation and long-term dynamic stability is indeed crucial. The results shown below will be integrated into the revised manuscript.
> To rigorously verify this capability, we conducted additional experiments by fine-tuning our pretrained model for short-term prediction (1 hour) to various extended lead times, including 2h, 3h, 4h, 5h, and 6h. This approach, which involves adjusting the temporal dimension within the TE block, follows the methodology described in Section 4.5 of our paper, with the newly added intermediate horizons (2h to 5h).
> We denote the fine-tuned models using the format "TE-Nf", where N represents the total number of predicted frames (e.g., "TE-12f" indicates a 2-hour prediction model, given a 10-minute frame interval). The CSI scores for these models across the six different prediction horizons are shown in the table below:
> | CSI-M (pool16) |   |      |      |      |     |  |
> |--|--|--|--|--|--|--|
> |  | 1h | 2h | 3h | 4h | 5h | 6h |
> |TE-6f| 0.4053 | -- | -- | -- | -- | -- |
> |TE-12f | 0.3770 | 0.2570 | -- | -- | -- | -- |
> |TE-18f | 0.3854 | 0.2585 | 0.1984 | -- | -- | -- |
> |TE-24f | 0.3508 | 0.2302 | 0.1745 | 0.1408 | -- | -- |
> |TE-30f | 0.3840 | 0.2783 | 0.2161 | 0.1752 | 0.1477 | -- |
> |TE-36f | 0.3923 | 0.2510 | 0.1773 | 0.1347 | 0.1075 | 0.0923 |
>
> The results demonstrate that by extending the predicted time horizon through the TE block, the model successfully achieves reliable and consistent accuracy across different lead times, without sacrificing the high accuracy achieved on the short-term horizon. This verification substantiates our claim regarding the flexibility and effectiveness of the TE block in adapting the model for various prediction lengths while maintaining forecast quality.

---

> ### Author Response · Authors · 2025-11-21
>
> **W3.  ... CSI as an indicator may not fully judge the quality of precipitation predictions ...**
> We appreciate the reviewer's insightful suggestions regarding the comprehensive assessment of precipitation prediction quality. Before addressing other potential metrics, we first wish to explain why the CSI is primarily considered in our study and is widely recognized as the most critical verification indicator for precipitation nowcasting:
> - Addressing Event Rarity and Data Imbalance: Traditional point-wise metrics like Mean Squared Error (MSE) and Mean Absolute Error (MAE) are inherently unreliable for this task because the scarcity of rare, high-intensity events (such as heavy or extreme rainfall) causes them to be averaged out by the dominant "no-rain" instances. In contrast, the CSI evaluates performance using a specific precipitation threshold, allowing it to isolate and accurately assess the prediction quality for these crucial rare events.
> - Balancing False Alarms and Misses: The CSI formula is robust because its denominator incorporates both “Misses” (failing to predict a real event) and “False Alarms” (predicting an event that didn't occur). This structure ensures that a high CSI score can only be achieved by a model that simultaneously minimizes both error types, providing a more comprehensive and reliable measure of forecasting skill than metrics focusing only on one error type.
> - Field Standardization and Interpretability: CSI is a long-established and iconic standard indicator within meteorological literature (see [1] for instance, the ECMWF Technical Memoranda No. 430). Its use enables direct and meaningful comparison with existing works and is highly interpretable by domain experts, confirming its status as the most appropriate metric for evaluating the practical skill of precipitation forecasts.
>
> With that being said, we agree with the reviewer that including additional indicators provides a more comprehensive assessment of the model's overall performance, particularly concerning the structure of the nowcasting outputs. To address this, we have additionally evaluated two distribution-based scores: the Fractional Skill Score (FSS) and the Regional Histogram Divergence (RHD), as proposed in [2]. These two metrics are specifically designed to assess the overall prediction distributions instead of pixel-wise values, and they effectively mitigate the "double penalty" issue that arises when counting Misses and False Alarms for slight spatial shifts in prediction.
> The comparison of our model and baselines on these two scores is shown in the table below:
>
> | Model            | Para (M) | Flops (G) | FSS     | RHD     |
> |------------------|----------|-----------|---------|---------|
> | UNet             | 19.3     | 28        | 0.4802  | 0.0478  |
> | ConvLSTM         | 16.6     | 7         | 0.5419  | 0.0366  |
> | PhyDNet          | 3.09     | 161       | 0.5264  | 0.0377  |
> | SimVP            | 1.9      | 42        | 0.5594  | 0.0325  |
> | Earthformer      | 15.1     | 61        | 0.5542  | 0.0352  |
> | AFNO             | 61.7     | 63        | 0.4332  | 0.0632  |
> | CasCast          | 391.0    | 1,729     | **_0.6108_**  | **0.0178**  |
> | AlphaPre         | 8.5      | 688       | 0.5720  | 0.0363  |
> | exPreCast (Ours) | 32.0     | 55        | **0.5950**  | **_0.0122_**  |
>
> We note that a higher FSS score indicates better distribution preservation (with 1 being perfect), while a lower RHD value signifies a smaller distribution distance between the prediction and ground truth (with 0 being perfect). Both metrics are evaluated using the same threshold as studied in the KMA dataset.
> The results further validate our model's superiority by demonstrating its ability to not only achieve high skill (CSI) but also to produce outputs with high structural fidelity (FSS) and accurate intensity distributions (RHD). It is worth to highlight again that although CasCast shows marginally better FSS score than ours, its computational overhead, quantified by GFLOPs, is more than 30 times greater than ours, depicting the computational burden of CasCast to achieve the marginal gain.
> - [1] Pertti Nurmi, Recommendations on the verification of local weather forecasts, ECMWF (1994)
> - [2] Yan et al, Fourier Amplitude and Correlation Loss: Beyond Using L2 Loss for Skillful Precipitation Nowcasting, NIPS (2024)

---

### Official Review · Reviewer_mKKA · 2025-11-01

**Soundness:** 3
**Presentation:** 3
**Contribution:** 3
**Rating:** 4
**Confidence:** 4

**Summary:**

The paper describes a deterministic method for generating radar forecasts. It also proposes a novel dataset, which is based on the data from Korea Meteorological Administration (KMA) and contains a balanced selection of meteorological events, which is aimed at reducing the bias towards either ordinary or extreme events. The choice of KMA data is due to the wide range of meteorological events: the country has intense rainfall in summer via monsoons and typhoons, and lighter precipitation in other seasons.

**Strengths:**

Originality:
- the contributions include both the dataset and the method for precipitation forecasting aimed specifically at enhancing extreme events nowcasting. The main methodological contribution includes the upsampling described in Section 3.2

Quality:
- Good performance: the experimental results (Table 1-3) show consistently good performance.

Clarity:
- the description of the work looks clear and easy to follow, and I believe the description is correct.

Significance:
- while the operational and novel deep-learning models cope reasonably well with the ordinary precipitation, the extreme precipitation is an important unsolved problem.
- availability of the big benchmarking dataset is an important asset for the research in the area, which also contributes towards the significance of this work.

**Weaknesses:**

Significance and originality:
- I can see there are two important points which are in advantage for the significance of the paper: (1) proposition of the Cubic Dual Upsample Block (2) dataset. Saying that, however, while the Cubic Dual Upsample Block is justified empirically in the ablation studies, it does not justify why it happens. Perhaps, one could create a link in the appendix, offering the analysis why this might lead to the improvements (if it follows from the existing literature such as Fan et al that would be also totally fine, however the justification needs to be there). This could be a theoretical justification or some additional empirical analysis which would help answer not only whether it helps but also what makes it work better. On point (2), the dataset, I would expect the authors to say whether it would be released (after careful reading I couldn't see it clearly), and put some descriptions about how this dataset is created. Now, I can only see in Appendix A that 'we first converted the radar reflectivity from dBZ to mm/h using the Marshall-Palmer Z-R relationship, Z = 200R1.6. We then report the CSI-p for thresholds p= [1,4,8,10,20,40,80]. Additionally, we applied a mask to exclude pixels outside the radar range' Maybe, comparing these statistics with the statistics of the existing datasets would be useful and would better justify this part of the contribution.

I would expect the authors also to put a list of contributions in the end of the intro, that would help navigate through the methodology.


Chi-Mao Fan, Tsung-Jung Liu, and Kuan-Hsien Liu (2022). SUNet: Swin transformer unet for image denoising.IEEE International Symposium on Circuits and Systems (ISCAS), pp. 2333–2337. IEEE, 2022

**Questions:**

1. Would it be possible to provide confidence intervals for Tables 1-3?

---

> ### Author Response · Authors · 2025-11-21
>
> **W1. ... why this (CDU) might lead to the improvements ...**
> We appreciate the reviewer’s suggestion for further clarification regarding the mechanism behind the effectiveness of the proposed CDU. We agree that a justification is crucial for supporting its benefits. While a purely theoretical justification on these modules remains challenging, we provide an empirical analysis based on the frequency decomposition and residual learning.
> The analysis assumes that the target image $Y$ can be decomposed into a smooth low-frequency component $Y_{low}$ and a detailed high-frequency (residual) component $Y_{high}$. That is,
> $Y = Y_{low​} + Y_{high}​$.
> First, traditional interpolation methods (e.g., Trilinear Interpolation) generate overly smooth results, where the underlying target image contains highly nonlinear behaviour between pixels. Therefore, the TI branch primarily approximates only the low-frequency content of the target image:
> $TI(X) \approx Y_{low}$
> On the other hand, while ​the Pixel Shuffle (PS) can capture the high-frequency part, it typically attempts to reconstruct the entire image in one step:
> $PS(X) \approx Y = Y_{low} + Y_{high}$
> However, it is empirically observed that forcing PS to learn both low- and high-frequency details simultaneously often leads to undesirable checkerboard artifacts in the output (as demonstrated in Figure 2 of our paper).
> Consequently, our proposed CDU, which fuses the TI and PS branches, leverages residual learning. By allowing the TI branch to capture the dominant $Y_{low}$​, the task of the PS branch is effectively decoupled. This enables the PS branch to focus primarily on reconstructing the high-frequency residual $Y_{high}$​. That is,
> $PS(X) → (Y - TI(X)) \approx Y_{high}$
> Therefore, by decoupling the reconstruction task, where the TI branch handles the smooth low-frequency content and the PS branch efficiently targets the high-frequency residuals, the CDU successfully mitigates the unrealistic artifacts while ensuring high structural fidelity in the final prediction. We will append those justification at the appendix in the revised version.
>
> **W2. ... whether it (KMA dataset) would be released ..., ... descriptions about how this dataset is created ...**
> We thank the reviewer for pointing out the unclear part in the description of the KMA dataset. The Korea Meteorological Administration (KMA) officially provides their product via their API hub, *https://apihub.kma.go.kr* and briefly explains their product in the document, *https://datawiki.kma.go.kr*. Among the products from KMA, we use Hybrid Surface Rainfall (HSR) composite radar images, which are generated at a 5-minute interval and cover $1152km\times 1440km$ with a spatial resolution of 500m.
> To construct the KMA dataset in this paper, we collect all images from 2014 to 2023 with a 10-minute interval, except some timestamps when no data were provided by KMA and downscale the spatial resolution as follows. First, we compute pixel coordinates $p_1$ located at $(lat,lon)=(29.0,120.5)$, and $p_2$ at $(lat,lon)=(42.0, 137.5)$ via Lambert conformal conic projection as described the KMA document. Then, we consider the rectangular area given by $p_1$ and $p_2$, i.e. the rectangle whose corner vertices are $p_1$ and $p_2$. We crop the central $1024km\times 1024km$ and subsample with stride 8 to make the spatial resolution $4km$, so the final sample in our dataset has size of $256\times256$ in the spatial axes. Lastly, we flip the image vertically to sort coordinate according to the latitude. This could be easily conducted by padding and slicing. We will revise the paper within the rebuttal period, to include those descriptions for the KMA.
> In addition, since KMA allows composite and secondary products to be openly shared provided that the source is clearly acknowledged, we plan to release both our dataset and the accompanying source code once the paper is published.

---

> ### Author Response · Authors · 2025-11-21
>
> **W3. ... comparing these statistics with the statistics of the existing datasets ...**
> We agree with the reviewer that a deeper examination of the statistical properties of the datasets provides invaluable insights. While the initial analysis and comparison of precipitation event distributions across datasets are already detailed in Appendix A.4 (Figures 8 and 9), we conducted further, more direct comparative analysis here.
> Specifically, we compared the KMA and MeteoNet datasets, as both are processed using the same standard ZR relationship, ensuring a fair comparison baseline. To conclusively clarify that KMA presents a balanced distribution while MeteoNet is biased towards normal rain, we conducted a statistical analysis of event probabilities based on an identical threshold set: [1,4,8,10,20,40,80] in mm/h units. The comparison is presented in the table below:
>
> | Threshold | Probability (KMA) | Probability (MeteoNet) |
> |-----------|--------------------|--------------------------|
> | 1 mm/h    | 0.18               | 0.22                     |
> | 4 mm/h    | 0.15               | 0.21                     |
> | 8 mm/h    | 0.05               | 0.09                     |
> | 10 mm/h   | 0.16               | 0.14                     |
> | 20 mm/h   | 0.17               | 0.14                     |
> | 40 mm/h   | 0.17               | 0.09                     |
> | 80 mm/h   | 0.13               | 0.11                     |
>
> The results clearly show that our KMA dataset exhibits more evenly distributed event probabilities across all specified thresholds compared to MeteoNet. Furthermore, when categorized into “light to normal” and “heavy to extreme”events, KMA contains nearly equal probabilities in both categories. In contrast, MeteoNet shows a pronounced bias towards the “light to normal” category. This analysis substantiates our claim that the KMA dataset offers a more balanced benchmark for precipitation nowcasting tasks.
>
> **W4. I would expect the authors also to put a list of contributions in the end of the intro, ...**
> Appreciating the reviewer’s comment, we will include the following list of contributions after revision.
> - We propose exPreCast, a deterministic radar-based forecasting architecture that achieves state-of-the-art accuracy across KMA, SEVIR, and MeteoNet, while maintaining substantially lower computational cost than diffusion-based counterparts.
> - We introduce the CDU block, which fuses trilinear interpolation with 3D pixel shuffle to mitigate checkerboard artifacts and retain small-scale high-intensity rainfall patterns. CDU consistently improves CSI scores, especially with pooling, demonstrating enhanced local-pattern fidelity.
> - We construct a new large-scale dataset encompassing both ordinary and extreme precipitation events, offering more balanced meteorological coverage than existing benchmarks and enabling rigorous evaluation of generalization.
>
> **Q1. Would it be possible to provide confidence intervals for Tables 1--3?**
> We appreciate the reviewer's suggestion concerning the confidence intervals. We agree that providing confidence intervals (CIs) offers valuable statistical insight into the stability and robustness of our model's performance. Due to strict time constraints, a full comparison of CIs across all baselines and datasets is not currently feasible. However, we provide here a preliminary stability analysis for our proposed model on the KMA dataset. We commit to completing a comprehensive CI study and integrating the full results into the revised manuscript.
> To thoroughly examine this stability issue, we verified our model's performance using a stratified K-fold method on the testing dataset. We chose this approach to avoid unbalanced event distribution across subsets, which is crucial due to the rarity of extreme precipitation events. We set K=5 subsets and evaluated the mean, standard deviation, and 95% confidence interval of the CSI scores. The results are presented in the table below:
>
> |||Mean$\pm$std|95% confidence interval|
> |--|--|--|--|
> |CSI-M|pool1|0.2794$\pm$0.0022|(0.2766, 0.2821)|
> ||pool4|0.3721$\pm$0.0018|(0.3698, 0.3744)|
> ||pool16|0.4841$\pm$0.0021|(0.4814, 0.4867)|
> |CSI-20|pool1|0.1786$\pm$0.0025|(0.1756, 0.1817)|
> ||pool4|0.3013$\pm$0.0024|(0.2983,0.3043)|
> ||pool16|0.4490$\pm$0.0030|(0.4453, 0.4527)|
> |CSI-80|pool1|0.0348$\pm$0.0016|(0.0329, 0.0367)|
> ||pool4|0.0835$\pm$0.0023|(0.0807, 0.0863)|
> ||pool16|0.1488$\pm$0.0027|(0.1454, 0.1522)|
>
> This analysis confirms that our proposed model is both stable and robust. Specifically, the model exhibits a low standard deviation across all rainfall types (as reflected by CSI-M), and especially for heavy (CSI-20​) and extreme (CSI-80​) cases, demonstrating the robustness of our model.

---

### Official Review · Reviewer_kH8J · 2025-11-09

**Soundness:** 3
**Presentation:** 3
**Contribution:** 3
**Rating:** 4
**Confidence:** 4

**Summary:**

This paper introduces exPreCast, a deterministic deep learning framework for extreme weather nowcasting using radar-based precipitation data.
The authors aim to address challenges in forecasting localized and fine-scale rainfall patterns, especially under extreme conditions, where existing diffusion-based generative models are accurate but computationally heavy, and deterministic ones are efficient but biased toward normal rainfall. exPreCast builds on a Video Swin Transformer backbone and incorporates two key modules: Cubic Dual Upsampling (CDU) and Temporal Extractor (TE). The authors also construct a new balanced radar dataset, KMA, covering both normal and extreme precipitation events in South Korea (2014–2023). Extensive ablation studies and long-term prediction results further validate the model’s robustness and efficiency.

**Strengths:**

Balanced new dataset: The introduction of the KMA dataset provides a valuable contribution for evaluating generalization across both normal and extreme rainfall conditions.

Strong empirical performance: exPreCast achieves SOTA results across multiple benchmarks, demonstrating robustness under diverse meteorological regimes.

Comprehensive experimentation: The paper includes rigorous comparisons, ablations, and qualitative visualizations that convincingly support the claims.

**Weaknesses:**

Lack of Discussion on FACL:
The authors employ FACL in their model but provide no related discussion. FACL contributes significantly to texture generation and substantial forecast improvements[1]. For fairness, the authors should present a comparison between other models (e.g., SIMVP) with FACL and the proposed exPreCast, or alternatively, compare the performance of exPreCast trained with MSE loss against other models.

Unfair Comparison Due to FACL in Different Forecast Durations:
Although the authors claim that the CDU decoder can adapt to various forecast lengths, the use of FACL still makes comparisons across different forecast durations unfair. It becomes difficult to distinguish whether the performance gain for long-term forecasting comes from FACL or from the CDU decoder itself.

Lack of Discussion on the Distribution of Precipitation Events Across Datasets:
Lines 18–21 state that previous datasets are limited to a single type of precipitation event, whereas KMA balances different precipitation types. However, this claim lacks statistical support. The authors should analyze and compare the distribution of precipitation events at different thresholds across datasets.

Incomplete Ablation Study on Upsampling:
The ablation on upsampling is only conducted on exPreCast, which is insufficient to demonstrate that upsampling is a general issue across all deterministic models. Moreover, the range of interpolation methods compared is limited—commonly used methods such as bicubic and area interpolation are missing. Exploring multiple interpolation approaches across different models would yield more valuable insights.

Incomplete Visualization in Figure 6:
Figure 6 should include all forecast frames from one hour to six hours after the first frame. Providing the complete sequence would allow reviewers to better evaluate the visual quality and temporal consistency of the generated forecasts.

Reference:
[1] Fourier Amplitude and Correlation Loss: Beyond Using L2 Loss for Skillful Precipitation Nowcasting [NIPS2024]

**Questions:**

How were the samples in the KMA dataset selected and quality-controlled?
What are the differences in MAE and MSE among the different models?

---

> ### Author Response · Authors · 2025-11-21
>
> **W1. Lack of Discussion on FACL**
> We sincerely appreciate the reviewer's thoughtful suggestion. We agree that the FACL loss function is indeed another crucial component for improving the performance of predictive texture and forecast quality. For a fair comparison, we have conducted an additional study, shown in the table below (and will be integrated into the Appendix of the revised manuscript), comparing our model with the top three performing baseline models trained with the adoption of FACL.
> | Model       | Loss      | CSI-M (pool1) | CSI-M (pool4) | CSI-M (pool16) | CSI-20 (pool1) | CSI-20 (pool4) | CSI-20 (pool16) | CSI-80 (pool1) | CSI-80 (pool4) | CSI-80 (pool16) | HSS    |
> |-------------|-----------|---------------|----------------|-----------------|----------------|----------------|------------------|----------------|----------------|------------------|--------|
> | SimVP       | -     | 0.2762        | 0.2558         | 0.2598          | 0.1536         | 0.1444         | 0.1612           | 0.0314         | 0.0381         | 0.0443           | 0.3958 |
> | | FACL        | 0.2691    | 0.3548        | 0.4717         | 0.1722          | 0.2890         | 0.4388         | 0.0282           | 0.0710         | 0.1305         | 0.3912           |        |
> | Earthformer | -    | 0.2518        | 0.2283         | 0.2302          | 0.1270         | 0.1146         | 0.1256           | 0.0084         | 0.0129         | 0.0165           | 0.3625 |
> | | FACL        | 0.1597    | 0.2712        | 0.3762         | 0.0519          | 0.1605         | 0.3121         | 0.0063           | 0.0332         | 0.0948         | 0.2417           |        |
> | AlphaPre    | -      | 0.2530        | 0.2311         | 0.2330          | 0.1210         | 0.1154         | 0.1299           | 0.0172         | 0.0204         | 0.0234           | 0.3533 |
> | | FACL        | 0.2539    | 0.3334        | 0.4367         | 0.1540          | 0.2549         | 0.3844         | 0.0232           | 0.0586         | 0.1000         | 0.3663           |        |
> | Ours | FACL | 0.2794    | 0.3721        | 0.4841         | 0.1786          | 0.3013         | 0.4490         | 0.0348           | 0.0835         | 0.1488         | 0.4042           |        |
>
> Here, the symbol '-' indicates the use of the respective baseline model's default loss function.
>
> The table shows that while FACL significantly improves the CSI scores with pooling for all baselines, our proposed model, exPreCast, consistently achieves the best performance. This superior result stems from the synergistic effects of our well-designed configuration, specifically the integration of the CDU, the TE block, and the use of the FACL loss function.
>
> **W2. Unfair Comparison Due to FACL in Different Forecast Durations**
> We would like to appreciate the insightful comments and agree that FACL may also contribute to performance improvements in long-term prediction. However, our main intention is not to attribute the adaptability to various forecast lengths to solely FACL or CDU, but rather to emphasize the crucial roles of the Temporal Extractor (TE) and the fine-tuning strategy.
> As shown in Table 5, whether the model is fine-tuned or not leads to a substantial performance gap in 6-hour forecasting. Moreover, by adjusting the TE block, we were able to obtain consistent performance across different lead times (both short-term and long-term) as shown in the following table. The TE module regulates the temporal resolution produced by the CDU decoder, and together with fine-tuning, it enables the model to reuse a single encoder while adapting effectively to multiple forecasting horizons.
>
> | CSI-M (pool16) |   |      |      |      |     |  |
> |--|--|--|--|--|--|--|
> |  | 1h | 2h | 3h | 4h | 5h | 6h |
> |TE-6f| 0.4053 | -- | -- | -- | -- | -- |
> |TE-12f | 0.3770 | 0.2570 | -- | -- | -- | -- |
> |TE-18f | 0.3854 | 0.2585 | 0.1984 | -- | -- | -- |
> |TE-24f | 0.3508 | 0.2302 | 0.1745 | 0.1408 | -- | -- |
> |TE-30f | 0.3840 | 0.2783 | 0.2161 | 0.1752 | 0.1477 | -- |
> |TE-36f | 0.3923 | 0.2510 | 0.1773 | 0.1347 | 0.1075 | 0.0923 |
>
> The results demonstrate that the ability to adapt to various forecast lengths and the observed performance gain are primarily driven by TE and the fine-tuning procedure, rather than by CDU or FACL themselves. We acknowledge that our original exposition may have created some ambiguity, and will revise the manuscript to clarify that TE and fine-tuning is the key in multi-horizon forecasting. We sincerely appreciate the reviewer’s helpful suggestion.

---

> ### Author Response · Authors · 2025-11-21
>
> **W3. Lack of Discussion on the Distribution of Precipitation Events Across Datasets**
> We appreciate the reviewer's insightful observation. We agree that the claimed diversity of the KMA dataset relative to existing benchmarks requires statistical validation. In fact, the statistical analysis and comparison of precipitation event distributions across datasets are already provided in Appendix A.4: Dataset Distribution and Comparison, specifically in Figures 8 and 9.
> Figure 8 presents a comprehensive event probabilities across various precipitation thresholds, which substantiates our claim:
> - The SEVIR dataset is significantly biased towards heavy rainfall events.
> - THe MeteoNet dataset leans towards normal rainfall events.
> - In contrast, our proposed KMA dataset exhibits a notably more balanced distribution across a broader spectrum of precipitation intensities, making it a more representative benchmark for diverse forecasting tasks.
>
> Furthermore, Figure 9 offers a detailed temporal view by visualizing the monthly histograms of different rainfall events in KMA and MeteoNet, further confirming the variability and richness of the KMA dataset.
>
> **W4. Incomplete Ablation Study on Upsampling**
> We sincerely appreciate the reviewer for raising this crucial point regarding the generality and completeness of the upsampling ablation study. We agree that examining the upsampling mechanism across different models and with a broader range of methods provides more robust insights.
> To thoroughly address this, we have significantly verified this claim in two ways:
> - *Investigating upsampling methods on a general model*
> To demonstrate that upsampling is a general issue across deterministic models, we conducted additional experiments on a plain UNet architecture. As the reviewer suggested, this study now includes a broader spectrum of upsampling methods in the decoder blocks: the previously studied Pixel Shuffle (PS) and Trilinear Interpolation (TRI), other interpolation approaches including Bicubic Interpolation (BIC) and Area Interpolation (AREA), and, in particular for this baseline, the commonly used method in traditional UNets: Transposed Convolution (ConvT).
> The results, presented in the table below, show that the proposed CDU consistently yields the best overall performance, as measured by CSI-M, among all other upsampling methods tested. Furthermore, the CDU maintains the highest CSI-M​ (1h) score, demonstrating its superior ability to preserve the structural fidelity of the prediction in the very last frame. Nevertheless, it is important to point out that, for all cases, the relatively lower scores on CSI-80​ and the nearly zero performance on CSI-80​ (1h) are due to the inherent limitations of the plain UNet architecture itself in handling extreme events.
> | UNet | CSI-M (pool1) | CSI-M (pool4) | CSI-M (pool16) | CSI-80 (pool1) | CSI-80 (pool4) | CSI-80 (pool16) | CSI-M (1h, pool1) | CSI-M (1h, pool4) | CSI-M (1h, pool16) | CSI-80 (1h, pool1) | CSI-80 (1h, pool4) | CSI-80 (1h, pool16) |
> |------------|---------------|----------------|-----------------|----------------|----------------|------------------|--------------------|--------------------|---------------------|----------------------|----------------------|-----------------------|
> | PS         | 0.2003        | 0.1781         | 0.1870          | 0.0067         | 0.0097         | 0.0130           | 0.1226             | 0.0983             | 0.0921              | 0.0000               | 0.0000               | 0.0000                |
> | TRI        | 0.1919        | 0.1644         | 0.1713          | 0.0037         | 0.0056         | 0.0078           | 0.1195             | 0.0907             | 0.0813              | 0.0000               | 0.0000               | 0.0000                |
> | BIC        | 0.1467        | 0.1309         | 0.1470          | 0.0031         | 0.0047         | 0.0065           | 0.0812             | 0.0622             | 0.0609              | 0.0000               | 0.0000               | 0.0000                |
> | AREA       | 0.1954        | 0.1708         | 0.1793          | 0.0064         | 0.0092         | 0.0121           | 0.1132             | 0.0980             | 0.0804              | 0.0000               | 0.0000               | 0.0000                |
> | ConvT      | 0.2028        | 0.1738         | 0.1784          | 0.0068         | 0.0092         | 0.0117           | 0.1141             | 0.0870             | 0.0790              | 0.0000               | 0.0000               | 0.0000                |
> | CDU        | 0.2224        | 0.1933         | 0.1923          | 0.0052         | 0.0070         | 0.0089           | 0.1396             | 0.1096             | 0.0951              | 0.0000               | 0.0000               | 0.0000                |

---

> ### Author Response · Authors · 2025-11-21
>
> - *Complete ablation study on upsampling methods in exPreCast*
> In addition to the generality study above, we also conducted a complete ablation study on our proposed model, exPreCast, comparing all these various upsampling methods (PS, TRI, BIC, AREA, and CDU).
> The results, shown in the table below, further confirm the effectiveness of our design:
> | exPreCast | CSI-M (pool1) | CSI-M (pool4) | CSI-M (pool16) | CSI-80 (pool1) | CSI-80 (pool4) | CSI-80 (pool16) | CSI-M (1h, pool1) | CSI-M (1h, pool4) | CSI-M (1h, pool16) | CSI-80 (1h, pool1) | CSI-80 (1h, pool4) | CSI-80 (1h, pool16) |
> |------------|---------------|----------------|-----------------|----------------|----------------|------------------|--------------------|--------------------|---------------------|----------------------|----------------------|-----------------------|
> | PS         | 0.2781        | 0.3577         | 0.4632          | 0.0372         | 0.0797         | 0.1379           | 0.1936             | 0.2577             | 0.3633              | 0.0075               | 0.0287               | 0.0771                |
> | TRI        | 0.2635        | 0.3564         | 0.4740          | 0.0295         | 0.0764         | 0.1436           | 0.1755             | 0.2530             | 0.3884              | 0.0076               | 0.0340               | 0.1023                |
> | BIC        | 0.2629        | 0.3616         | 0.4814          | 0.0307         | 0.0830         | 0.1530           | 0.1785             | 0.2599             | 0.3937              | 0.0098               | 0.0427               | 0.1156                |
> | AREA       | 0.2631        | 0.3584         | 0.4765          | 0.0305         | 0.0773         | 0.1466           | 0.1753             | 0.2531             | 0.3879              | 0.0071               | 0.0345               | 0.1053                |
> | CDU        | 0.2794        | 0.3721         | 0.4841          | 0.0348         | 0.0835         | 0.1488           | 0.1963             | 0.2772             | 0.4053              | 0.0116               | 0.0454               | 0.1197                |
>
> Once again, exPreCast, featuring the intricately configured CDU module, consistently yields superior performance, demonstrating the effectiveness and utility of the proposed CDU as a robust upsampling method.
>
> **W5. Incomplete Visualization in Figure 6**
> We appreciate the reviewer's insightful observation regarding the visualization of the complete forecast sequence. Due to the page limit, the full six-hour prediction sequence for all compared baseline models is already provided in Appendix D.4, Figure 20.
> This comprehensive figure illustrates the prediction evolution from the first hour up to six hours ahead. We clearly observe that as the prediction horizon extends, the forecasts generated by baseline models progressively degrade and become overly blurred, reflecting a loss of spatial detail and temporal fidelity. In contrast, our proposed model, exPreCast, consistently maintains a fine-grained structural coherence throughout the entire six-hour prediction period. This qualitative superiority demonstrates that exPreCast not only achieves quantitatively better scores but also delivers significantly higher qualitative performance in maintaining the crucial details required for reliable long-term forecasting.

---

> ### Author Response · Authors · 2025-11-21
>
> **Q1. How were the samples in the KMA dataset selected and quality-controlled?**
> The KMA dataset used in the paper is not manually processed or quality-controlled by us. Instead, we simply collect the radar images publicly provided by the Korea Meteorological Administration (KMA) through *https://apihub.kma.go.kr*. KMA briefly explains their radar processing and QC procedures, as well as the types of radar products they release, in their official documentation at *https://datawiki.kma.go.kr*. We will clarify this in the revised version.
> For this study, we constructed the KMA dataset by downloading the Hybrid Surface Rainfall (HSR) composite radar images from 2014 to 2023. The products are provided at 5-minute intervals and cover $1152km\times1440km$ at $500m$ resolution, consisting of $2305\times 2881$ pixels, which are downsampled through preprocessing described in the next paragraph. We downloaded all available images at a 10-minute interval, except some timestamps just because no data were available from KMA at those times. We did not remove any samples by any additional filters.
> For details on the downsampling, we first use Lambert conformal conic projection to compute the area range from latitude $29.0^\circ$ to $42.0^\circ$ and longitude $120.5^\circ$ to $137.5^\circ$. Then, we crop the central $1024km\times1024km$ of the area and uniformly subsample with stride 8 to generate $256\times256$ images with the spatial resolution of $4km$. Lastly, we vertically flip to ensure north-south ordering aligned with increasing latitude. As KMA affirmed that the constructed dataset could be open to the public when properly attributed, we will share those dataset as well as the source code.
> Although the quality of radar products heavily rely on the KMA itself, KMA serves as South Korea’s official government meteorological authority and participates in the World Meteorological Organization (WMO). Hence, we regard the dataset as highly reliable as KMA and WMO.
>
> **Q2. What are the differences in MAE and MSE among the different models?**
> We appreciate the reviewer for requesting additional evaluation using metrics such as MAE and MSE. We are pleased to provide the comparison between our model and the baselines on these metrics, as shown in the table below:
>
> | Model            | MAE     | MSE      |
> |------------------|---------|----------|
> | UNet             | 0.0753  | 10.6032  |
> | ConvLSTM         | 0.0524  | 0.5755   |
> | PhyDNet          | 0.0488  | 0.5474   |
> | SimVP            | 0.0482  | 0.5219   |
> | Earthformer      | 0.0514  | 0.5579   |
> | AFNO             | 0.0622  | 0.7400   |
> | CasCast          | 0.0958  | 3.1821   |
> | AlphaPre         | 0.0491  | 0.5420   |
> | exPreCast (Ours) | 0.0639  | 3.1342   |
>
> Our model achieves comparable accuracy to most baseline models on these point-wise metrics, which demonstrates our reliable ability for general image reconstruction. It is important to note that while our model exhibits a relatively higher MSE than some baselines, these competing models were trained directly using the MSE loss function, which inherently biases their results towards a lower score on this specific metric.
> Nevertheless, it is crucial to note that point-wise metrics such as MAE, MSE, or other pixel-based quantities (e.g., SSIM, PSNR) are generally considered insufficient for robustly evaluating precipitation forecasting ability (see [1] for instance, the ECMWF Technical Memoranda No. 430). As widely recognized in existing meteorological literature, MSE, for instance, tends to smooth out high-intensity regions (where heavy rainfall events often occur), thereby leading to a significant underestimation of performance in extreme cases.
> Therefore, the CSI scores are primarily considered in meteorological literature, as well as in our paper, because it effectively captures prediction performance across various event intensity levels.
>
> [1] Pertti Nurmi, Recommendations on the verification of local weather forecasts, ECMWF (1994)

---

### Author Response · Authors · 2025-11-28

Thank you very much for your thoughtful and constructive feedback on our submission. We sincerely appreciate the time you have taken to evaluate our work. We have carefully addressed all comments and prepared detailed responses to each question. We hope that our clarifications fully resolve your concerns, and we would be happy to provide further explanations or additional experiments if needed.
During the rebuttal period, we also conducted new experiments inspired by the reviewers’ questions, which further strengthen the empirical validation of our method. All newly added or revised content has been incorporated into the updated manuscript, with changes highlighted in blue for clarity.
A summary of the newly added experiments is as follows:
- compare CDU to other upsampling methods such as Bicubic and Area interpolations.
- comparison of TE modules for different predicted time horizons, from 1-hour to 6-hours.
- compare to baselines trained via FACL, instead of loss provided in their original paper.
- Evaluate MAE, MSE, FSS, and RHD on KMA dataset.

Thank you again for your time and effort in reviewing our paper. Please feel free to let us know if you have any additional questions. We would be glad to assist.

---

### Meta-Review · Area_Chair_daaU · 2025-12-29

**Summary:**

Provide a summary of the reviewers' concerns that informed your suggested decision for this paper.

All reviewers recognized the paper’s two main contributions: (1) the Cubic Dual-Upsample (CDU) decoder for improving texture fidelity and (2) the Korea Meteorological Administration (KMA) dataset. Reviewers also consistently noted that exPreCast achieves strong performance across settings.

Overall, the rebuttal addressed most of the raised concerns.

- For kH8J, the authors provided additional ablations (including comparisons between FACL and MSE losses and among different upsampling methods) and clarified the data distribution details in Appendix A.4.
- For mKKA, the rebuttal improved the theoretical motivation for CDU and added details on the dataset creation process.
- For Fimn, the authors included new comparisons against GAN-based methods, added results analyzing the Temporal Extractor (TE block), and reported additional metrics beyond CSI, including Fractional Skill Score (FSS) and Regional Histogram Divergence (RHD).
- For aEjv, the authors provided a per-module contribution table and expanded the description of the KMA dataset.

Given that no major issues appear to remain unaddressed and the paper makes solid contributions through both the CDU decoder and the KMA dataset, I recommend acceptance.

**Reviewer Concerns:**

The author addressed most of the concerns.

- For kH8J, the authors provided additional ablations (including comparisons between FACL and MSE losses and among different upsampling methods) and clarified the data distribution details in Appendix A.4.
- For mKKA, the rebuttal improved the theoretical motivation for CDU and added details on the dataset creation process. Whether the new theoretical motivation addressed the concern is subjective.
- For Fimn, the authors included new comparisons against GAN-based methods, added results analyzing the Temporal Extractor (TE block), and reported additional metrics beyond CSI, including Fractional Skill Score (FSS) and Regional Histogram Divergence (RHD).
- For aEjv, the authors provided a per-module contribution table and expanded the description of the KMA dataset.

**Reviewer Scores:**

I think kH8J may very likely increase the score because the author conducted extensive new experiments to address the concerns.

For mKKA, I believe the score will be raised after discussing with the other reviewers.

Fimn and aEjv may also increase the score since the author answered their questions with new results.

---

### Decision · Program_Chairs · 2026-01-26

Accept (Poster)